# Spatially informed clustering, integration, and deconvolution of spatial transcriptomics with GraphST

Yahui Long[1], Kok Siong Ang[1], Mengwei Li[1], Kian Long Kelvin Chong[1], Raman Sethi[1], Chengwei Zhong[1], Hang Xu[1], Zhiwei Ong [2], Karishma Sachaphibulkij[3,4], Ao Chen [5,6,7], Li Zeng [2,8], Huazhu Fu [9], Min Wu[10], Lina Hsiu Kim Lim[3,4], Longqi Liu [7] & Jinmiao Chen [1,3] ✉

Spatial transcriptomics technologies generate gene expression profiles with spatial context, requiring spatially informed analysis tools for three key tasks, spatial clustering, multisample integration, and cell-type deconvolution. We present GraphST, a graph self-supervised contrastive learning method that fully exploits spatial transcriptomics data to outperform existing methods. It combines graph neural networks with self-supervised contrastive learning to learn informative and discriminative spot representations by minimizing the embedding distance between spatially adjacent spots and vice versa. We demonstrated GraphST on multiple tissue types and technology platforms. GraphST achieved 10% higher clustering accuracy and better delineated fine-grained tissue structures in brain and embryo tissues. GraphST is also the only method that can jointly analyze multiple tissue slices in vertical or horizontal integration while correcting batch effects. Lastly, GraphST demonstrated superior cell-type deconvolution to capture spatial niches like lymph node germinal centers and exhausted tumor infiltrating T cells in breast tumor tissue.

Within the tissues of multicellular organisms, cells are organized into groups of similar cells physically clustered together. Linking gene expression of cells with their spatial distribution is crucial for understanding the tissue's emergent properties and pathology[1]. Using spatial transcriptomics (ST), we can concurrently capture gene expression profiles and spatial information to achieve greater insights into both healthy and diseased tissues[2–4]. Spatial information is also useful for inferring cell–cell communications, especially juxtacrine signaling[5]. Within the spatial transcriptomics analysis workflow, assigning capture spots to spatial domains with unsupervised clustering is an essential task. Among the existing clustering methods employed in spatial domain identification, k-means, Louvain's method[6], and Seurat[7] utilize only gene expression data to cluster spots into different domains. The domains identified by these methods are often discontinuous as they

[1]Singapore Immunology Network (SIgN), Agency for Science, Technology and Research (A*STAR), 8A Biomedical Grove, Immunos Building, Level 3, Singapore 138648, Singapore. [2]Neural Stem Cell Research Lab, Research Department, National Neuroscience Institute, 11 Jalan Tan Tock Seng, Singapore 308433, Singapore. [3]Immunology Translational Research Program, Department of Microbiology and Immunology, Yong Loo Lin School of Medicine, National University of Singapore (NUS), 5 Science Drive 2, Blk MD4, Level 3, Singapore 117545, Singapore. [4]Department of Physiology, Yong Loo Lin School of Medicine, NUS, 2 Medical Drive, MD9, Singapore 117593, Singapore. [5]BGI Research-Southwest, BGI, 401329 Chongqing, China. [6]JFL-BGI STOmics Center, Jinfeng Laboratory, 401329 Chongqing, China. [7]BGI Research-ShenZhen, BGI, 518083 Shenzhen, China. [8]Neuroscience and Behavioral Disorders Program, DUKE-NUS Graduate Medical School, Singapore 169857, Singapore. [9]Institute of High Performance Computing (IHPC), Agency for Science, Technology and Research (A*STAR), 1 Fusionopolis Way, #16-16 Connexis, Singapore 138632, Singapore. [10]Institute for Infocomm Research (I2R), Agency for Science, Technology and Research (A*STAR), 1 Fusionopolis Way, #21-01 Connexis, Singapore 138632, Singapore. ✉e-mail: chen_jinmiao@immunol.a-star.edu.sg

do not employ spatial information to identify colocalized cells that are likely to belong to the same domain.

Recently, several methods have been proposed to improve spatial domain identification by exploiting spatial information. For example, Giotto[8] uses a Hidden Markov Random Field (HMRF) model to detect spatial domains with coherent gene expression patterns by fully exploiting the spatial dependency between spots. SpaGCN[9] uses a graph convolutional network-based model to identify spatial domains by integrating gene expression, spatial location, and histology image. stLearn[10] combines morphological features obtained from the histology image with gene expression of adjacent spots to cluster similar spots in the tissue. BayesSpace[11] adopts a Bayesian statistical method that employs the information from spatial neighborhoods to improve clustering analysis. More recently, STAGATE[12] uses a graph attention auto-encoder framework to identify spatial domains by integrating spatial information and gene expression profiles. Another method, Cell Clustering for Spatial Transcriptomics data (CCST), uses a graph convolutional network for unsupervised cell clustering[13]. However, these methods employ unsupervised learning, and they often show suboptimal clustering performance as the boundaries of identified domains are often fragmented and poorly match the pathological annotations. As ST ground-truth segmentation is usually not available, supervised learning cannot be employed to improve performance. Alternatively, self-supervised learning that uses unlabeled input data has been applied to identify spatial domains. For example, SpaceFlow[14] employs a deep graph neural network with a contrastive learning strategy that creates negative examples by randomly permuting its spatial expression graph during the encoder learning process. conST[15] proposes a contrastive learning framework with two training stages to learn a representative low-dimensional embedding. The second stage uses contrastive learning at three levels of contrast. However, the lack of consideration of the spots' local context hampers their performance.

Spatial transcriptomics technologies have size restrictions on the area captured during data acquisition. To perform spatial transcriptomics on a tissue slice encompassing a whole organ of interest, the sample is dissected into multiple sections. These adjacent multiple sections will have to be inferred jointly to accurately identify tissue compartments within the whole organ. Multiple tissue sections can also be obtained via serial sectioning of the organ of interest, yielding serial tissue slices that capture 3D information in each spatial transcriptomics experiment. Therefore, there is a need for methods to integrate and learn the joint representation of adjacent tissue sections (horizontal integration) and serial tissue sections (vertical integration). Most current analysis methods are suitable for only single tissue slices and cannot jointly identify spatial domains from multiple slices. Moreover, batch correction methods developed for scRNA-seq, such as Harmony[16] and scVI[17], are not suitable as they only consider gene expression and do not employ the associated spatial information. Although STAGATE can be used to analyze multiple tissue slices, its performance is limited by the lack of batch effect removal capabilities. As such, spatially informed batch correction tools are needed for ST data.

Current technological limitations also prevent ST from achieving single-cell resolution with gene coverage comparable to single-cell RNA sequencing (scRNA-seq). The popular 10x Visium platform can capture scRNA-seq scale transcriptomes but uses 55 μm capture spots that are larger than typical cells (5–10 μm)[18,19]. More recent sequencing-based technologies, including Slide-seq[20], DBiT-seq[21], Stereo-seq[22], PIXEL-seq[23], and Seq-Scope[24], offer subcellular spatial resolution but suffer from high dropout events that give rise to a very sparse gene-spot matrix. Meanwhile, fluorescent in situ hybridization (FISH)-based methods are capable of achieving subcellular resolution but lack genome-scale gene coverage, with the latest seqFish having a 10,000 gene limit[25]. To analyze data from low-resolution capture techniques,

computational methods such as RCTD[26], stereoscope[27], SPOTlight[28], cell2location[29], CARD[30], NNLS (AutoGeneS)[31,32], and spatialDWLS[33] have been developed. These methods perform cell-type deconvolution of low-resolution spots by leveraging cell-type-specific gene expression from RNA-seq. However, all current deconvolution methods except CARD ignore spatial location information. Moreover, they only return a matrix of cell-type composition, except RCTD and cell2location, which also calculate the cell-type-specific gene expressions for each location. Currently, cell-level deconvolution methods for achieving single-cell resolution of spatial transcriptomes are still lacking. Alternative to deconvolution, the projection of scRNA-seq data onto ST data can create single-cell resolution spatial transcriptomic maps with genome-wide gene coverage, which can also be used to infer cell-type compositions of ST spots and spatial localization of scRNA-seq. Tangram[34] and other similar packages adopt a batch integration approach to correct for technical variations in the scRNA-seq and ST data to accomplish projection but do not employ spatial information. Therefore, spatially informed integration of scRNA-seq data with ST is needed to achieve more accurate cell-type deconvolution of ST and spatial registration of scRNA-seq.

To address the aforementioned challenges, we developed GraphST, a graph self-supervised contrastive learning method that makes full use of spatial information and gene expression profiles for spatially informed clustering, batch integration, and cell-type deconvolution. By using self-supervised contrastive learning in GraphST, we found that it improves performance in learning relevant latent features for downstream analysis. GraphST first combines graph neural networks with augmentation-based self-supervised contrastive learning to learn representations of spots for spatial clustering by encoding both gene expression and spatial proximity. For the cell-type deconvolution task, we train an auto-encoder to learn informative cellular features from the scRNA-seq data in an unsupervised way. Thereafter, GraphST learns a mapping matrix to project the scRNA-seq data into the ST space based on their learned features via an augmentation-free contrastive learning mechanism where the similarities of spatially neighboring spots are maximized, and those of spatially non-neighboring spots are minimized. The mapping matrix is then used to estimate the cell-type compositions of ST spots. We extensively tested GraphST for the three analysis tasks on different 10x Visium, Stereo-seq, and Slide-seqV2 datasets of human and mice tissues, including the human brain, human breast cancer tissue, human lymph node, mouse breast cancer, mouse olfactory bulb, mouse brain, and mouse embryo. The clustering tests demonstrated GraphST's superiority over seven existing methods in identifying spatial domains. Joint analyses on the mouse breast cancer and mouse brain datasets showed that GraphST was able to accurately identify spatial domains from multiple tissue slices while removing batch effects effectively without needing to explicitly detect batch factors. We also tested GraphST's projection of scRNA-seq data onto ST to predict cell states (cell types and sample types) in spatial spots. The computed cell-to-spot mapping matrix gave a more accurate estimate of cell-type compositions than cell2location, the best-performing deconvolution method. Moreover, GraphST can transfer scRNA-seq-derived sample phenotypes onto ST. We demonstrated this capability by delineating tumor and normal adjacent regions in a tumor-derived tissue slice.

## Results
### Overview of GraphST
GraphST comprises three modules, each with graph self-supervised contrastive learning architectures tailored for the three tasks respectively: spatially informed clustering (Fig. 1A), vertical and horizontal batch integration of multiple tissue sections (Fig. 1B), and spatially informed cell-type deconvolution by projecting scRNA-seq to ST (Fig. 1C). In all three modules we leverage the spatial information of the spatial transcriptomics dataset to construct a neighborhood graph

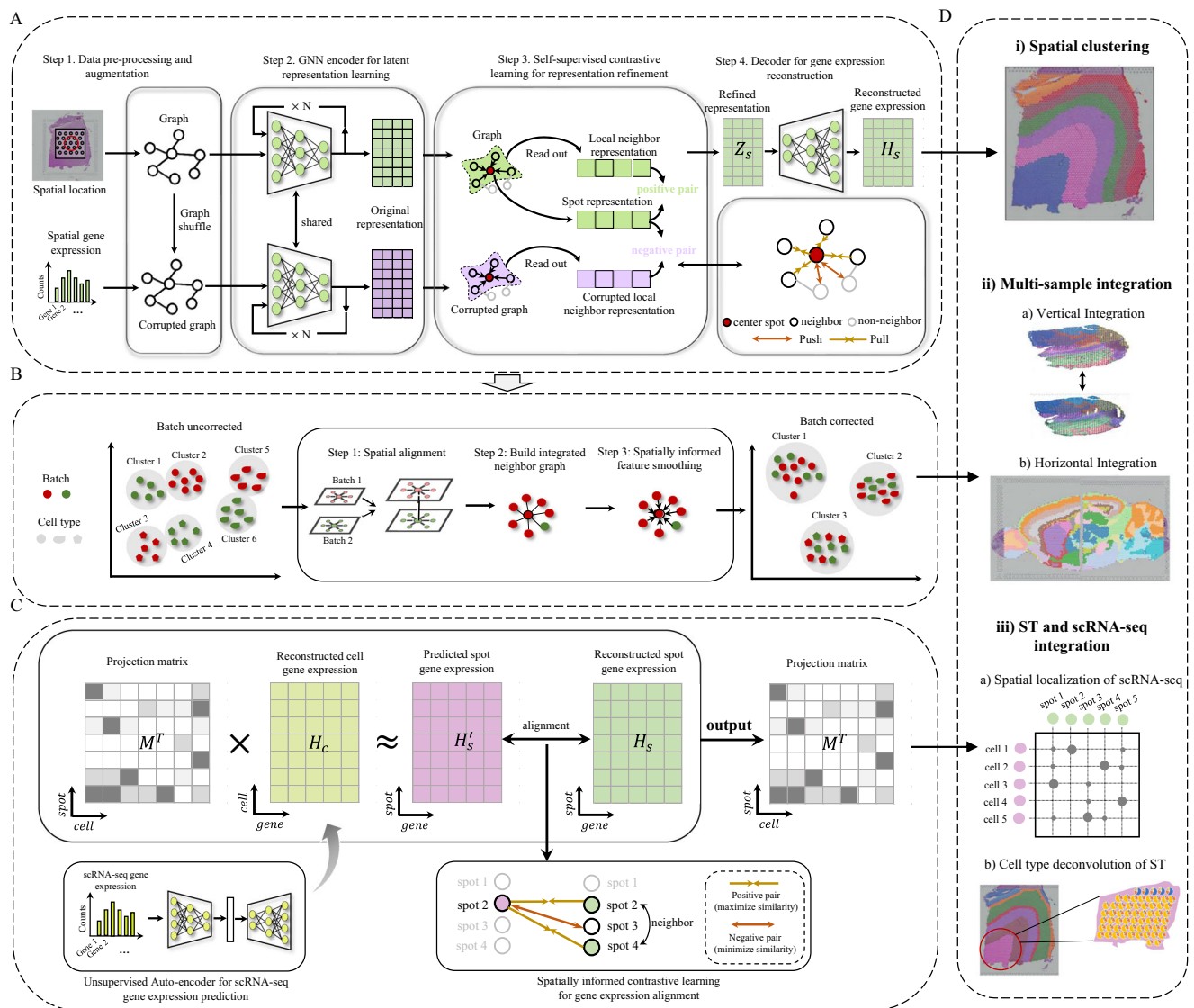

**Fig. 1 | Overview of GraphST. A** GraphST takes as inputs the preprocessed spatial gene expressions and neighborhood graph constructed using spot coordinates $(x, y)$. Latent representation $Z_s$ is first learned using our graph self-supervised contrastive learning to preserve the informative features from the gene expression profiles, spatial location information, and local context information. This is then reversed back into the original feature space to reconstruct the gene expression matrix $H_s$. **B** The analysis workflow for spatial batch effect correction by GraphST. The first step is to align the H&E images of two or more samples, followed by shared neighborhood graph construction, where both intra- and inter-sample neighbors are considered. This provides the possibility for feature smoothing. Finally, sample batch effects are implicitly corrected by smoothing features across samples with GraphST. **C** With the reconstructed spatial gene expression $H_s$ and the refined scRNA-seq feature matrix $H_c$ derived from an unsupervised auto-encoder, a cell-to-spot mapping matrix $M$ is trained via a spatially informed contrastive learning mechanism where the similarities of positive pairs (i.e., spatially adjacent spot pairs) are maximized, and those of negative pairs (i.e., spatially nonadjacent spot pairs) are minimized. **D** The outputs $H_s$ and $M$ of GraphST can be utilized for spatial clustering, multiple ST data integration, and ST and scRNA-seq data integration.

where spots spatially close to each other are connected. Next, a graph convolutional network is built as an encoder to embed the gene expression profiles and spatial similarity into a latent representation space by iteratively aggregating gene expressions from neighboring spots.

In the spatial clustering module, we adopt augmentation-based contrastive learning. We augment data by creating a corrupted graph by randomly shuffling gene expression vectors across spots while keeping the adjacency matrix of the original graph unchanged. As is observed in many tissue types, a spot in the spatial data usually contains cell types similar to its local context, e.g., one-hop or two-hop neighbors. Different spots in a tissue sample can have different local spatial contexts. Compared to a global context, the local context can better preserve spot-specific local neighborhood information and spot-to-spot variability. Therefore, our self-supervised contrastive

learning strategy enforces spot embedding to capture local context, while most other contrastive learning methods capture global context. Specifically, we define positive pairs by pairing spot embedding with its local summary vector and negative pairs by pairing the local summary vector with its corresponding spot embedding from the corrupted graph. The local summary vector represents the local context of a spot and is obtained by a sigmoid of the mean of all its neighbors' embeddings. This process brings spot embedding closer to its real local context from the original graph and pushes it away from its fake local context from the corrupted graph, such that spatially adjacent spots will have similar embeddings while nonadjacent spots will have dissimilar embeddings. Furthermore, considering that the corrupted graph has the same topological structure as the original graph, we define a corrupted contrastive loss (formula (5)) for the corrupted graph and combine it with the original contrastive loss (formula (4)) to

form a symmetric contrastive loss, which can make the model training more stable and balanced. In addition to the contrastive loss, GraphST's objective function includes a self-reconstruction loss. The learned representation $Z_s$ is reversed back into the original space through a decoder to obtain the reconstructed gene expression matrix $H_s$. The learning process jointly optimizes for the self-reconstruction loss and contrastive loss. The self-reconstruction loss enforces $H_s$ to retain the key feature information contained in the gene expression, while the contrastive loss encourages the gene expression matrix $H_s$ to capture the spatial context information.

In the vertical and horizontal integration module, GraphST learns batch-corrected representations of spots that can be used for downstream analysis, such as joint spatial clustering to map common spatial domains across multiple tissue sections. Before training the model in the GraphST framework, multiple sections are aligned vertically or stitched horizontally, and a shared graph is constructed. Thereafter, graph self-supervised contrastive learning is performed using the shared graph, which allows for the GraphST model to automatically smooth features of adjacent spots within and across sections. Therefore, after model training, not only do neighboring spots from the same section have similar representations, but neighboring spots from different sections will also have similar representations. Our GraphST framework removes batch effects implicitly during model training, unlike most existing approaches that remove batch effects by explicitly detecting batch factors, such as Harmony[16], the mutual nearest neighbors (MNN) approach[35], and Seurat 3.0[8]. Moreover, as these methods were designed for nonspatial scRNA-seq data integration, they ignore spatial information when correcting for batch effects.

In the third module, GraphST projects scRNA-seq to ST by augmentation-free contrastive learning to achieve spatially informed cell-type deconvolution of ST spots. To reduce noises resulting from sequencing technology, we use an auto-encoder to learn cell representation $H_c$ from the scRNA-seq gene expression. With the cell representation $H_c$ and the spot representation $H_s$ learned from the first module, we project the scRNA-seq data onto the spatial transcriptome data via a trainable mapping matrix $M$, which denotes the probability of the cells being projected onto each spot of the spatial data. This process is implemented by aligning the predicted spatial gene expression $H_s' = M^T \cdot H_c$ to the reconstructed spatial gene expression $H_s$ via an augmentation-free contrastive learning mechanism, where the similarities of positive pairs (i.e., spot $i$ and its spatial neighbors) are maximized while those of negative pairs (i.e., spot $i$ and its spatial non-neighbors) are minimized. During model training, the first and third modules are independently executed. After model training, $H_s$ and $M$ can be applied for various tasks, such as spatial clustering, multiple ST data integration, and scRNA-seq and ST data integration.

## GraphST's spatial clustering of human dorsolateral prefrontal cortex 10x Visium data improved the identification of known layers

We first assessed GraphST's spatial clustering performance on the LIBD human dorsolateral prefrontal cortex (DLPFC) dataset[36]. This dataset contains spatially resolved transcriptomic profiles of 12 DLPFC slices, each depicting the four or six layers of the human dorsolateral prefrontal cortex and white matter (WM). Here we tested GraphST and competing methods for their ability to recover the annotated anatomical cortex layers in an unsupervised manner. Across all 12 slices, GraphST achieved the highest median score of 0.60 among all methods, with only STAGATE's performance as a close second (Fig. 2A). GraphST also achieved lower variability in performance across the slices than most of the other methods. The remaining methods had median ARI scores of less than 0.53, and for some methods like SpaGCN, SpaceFlow, and STAGATE, there was a wide variance in performance across different slices. Seurat and conST achieved low

variance in their ARI scores, similar to GraphST, but their median scores were worse.

Next, we illustrate the results with one slice (#151673) in Fig. 2B–C. The visual comparison clearly showed that the Seurat clustering had the poorest performance. It was only able to recover the Layer 1 clustering with the remaining clusters mixed among the other layers, including the WM being clustered together with a portion of Layer 6. The boundaries between clusters were also ragged with no clean separation. Giotto and SpaceFlow accomplished better separation of WM and Layer 1, but Layers 2–6 were not correctly recovered. conST identified well-defined layers, but most of the layers were inconsistent with the annotation. SpaGCN, BayesSpace, and STAGATE produced layers that were closer in shape to the annotation but with incorrect layer thickness. The methods also did not accurately capture the boundary between Layer 6 and WM, which was accomplished by GraphST, the only method that could do so. For quantitative assessment, we employed the widely used Adjusted Rand Index (ARI). GraphST achieved the highest score of 0.64, while Seurat was the poorest at 0.29, followed by Giotto at 0.34. The remaining methods (SpaGCN, BayesSpace, SpaceFlow, conST, STAGATE) obtained similar scores between 0.40 and 0.58. The results with all other slices are in Supplementary Fig. S1.

Among the methods tested in this example, we only applied Seurat, STAGATE, and GraphST to the subsequent Stereo-seq and Slide-seq examples due to data format incompatibility. Consequently, we also tested all eight methods on a 10x Visium mouse brain dataset[19] and showed that GraphST was the best method. The full analysis results are available in Supplementary File 1.

## GraphST's spatial clustering of mouse olfactory bulb Stereo-seq data better demarcated the laminar structure

In this second example, we used a coronal mouse olfactory bulb tissue dataset acquired with Stereo-seq[22]. We first annotated the coronal mouse olfactory bulb's laminar structure using the DAPI-stained image, identifying the olfactory nerve layer (ONL), glomerular layer (GL), external plexiform layer (EPL), mitral cell layer (MCL), internal plexiform layer (IPL), granule cell layer (GCL), and rostral migratory stream (RMS) (Fig. 2D). Overall, all three tested methods (Seurat, STAGATE, and GraphST) were able to separate the outer layers of the organ, namely the ONL, GL, and EPL (Fig. 2E). The results of Seurat were slightly poorer with greater mixing between clusters, which we attribute to its lack of consideration of spatial information. For the inner structure, only GraphST was able to demarcate the GCL and RMS regions. Seurat was able to find the RMS but merged the GCL with the outer IPL region. STAGATE was unable to capture the laminar structure for effective identification of the relevant inner layers, particularly the RMS, GCL, and IPL regions. We next used the respective marker genes of each anatomical region to validate GraphST's results (Fig. 2F). Here, we see good correspondence between GraphST's clusters and the known marker genes. For some marker genes, such as *Mbp* and *Pcp4*, their high expression levels overlapped with neighboring regions. This is expected as cell types are often shared among the different inner structures of organs, and markers are likewise shared among similar cell types. Overall, GraphST was able to leverage the whole transcriptome and spatial information to discern the relevant anatomical regions.

## GraphST's spatial clustering of mouse hippocampus Slide-seqV2 data more accurately discerned the relevant anatomical regions

In this example, we compared Seurat, STAGATE, and GraphST using a mouse hippocampus dataset acquired with Slide-seqV2. For this comparison, we employed the annotated Allen Brain Atlas as the ground truth (Fig. 2G). Although Seurat was able to outline the major anatomical regions, many clusters were intermixed (Fig. 2H). We hypothesize that this was a result of Seurat capturing different cell

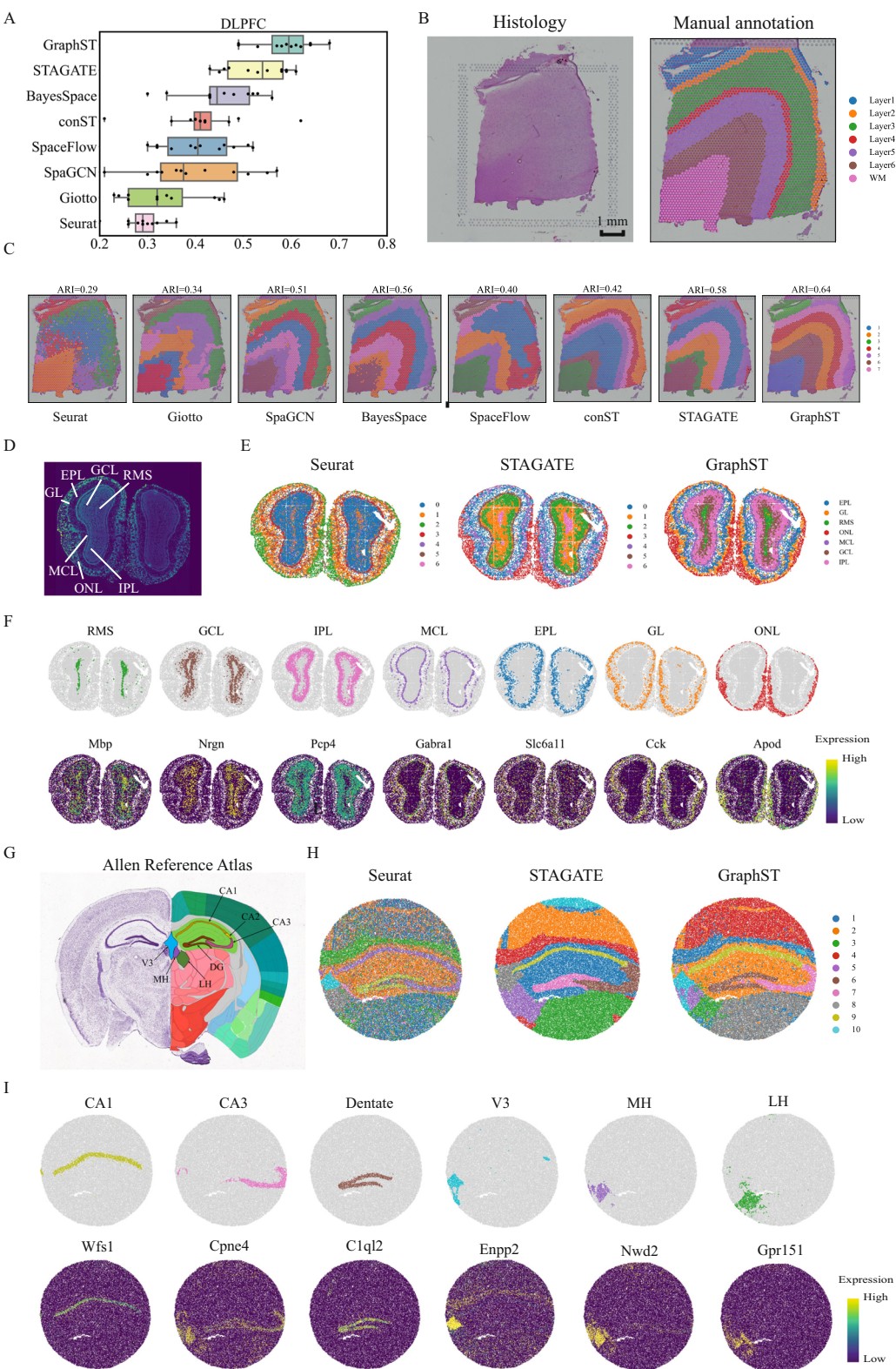

types that reside across different anatomical regions; this highlights the importance of local spatial information in achieving anatomically relevant clustering. STAGATE and GraphST produced more spatially consistent clustering and captured major anatomical regions such as the dentate gyrus (DG) and the pyramidal layers within Ammon's horn, which can be further separated into fields CA1, CA2, and CA3. Here, GraphST was better than STAGATE in delineating the CA3 and DG regions with sharper boundaries and higher concordance with the

anatomical annotation (Fig. 2G) and marker gene expressions (Fig. 2I). Unlike the other two methods, GraphST was able to differentiate between the third ventricle (V3), medial habenula (MH), and lateral habenula (LH). In contrast, Seurat merged the LH with the rest of the thalamus, while STAGATE merged the MH and LH into one region.

We further examined the regions' marker gene expressions to verify GraphST's clustering (Fig. 2I). For most regions and their corresponding markers, they showed good alignment. In particular, only

**Fig. 2 | GraphST clustering improves the identification of tissue structures in the human dorsolateral prefrontal cortex (DLPFC), mouse olfactory bulb, and mouse hippocampus tissue. A** Boxplots of adjusted rand index (ARI) scores of the eight methods applied to the 12 DLPFC slices. In the boxplot, the center line denotes the median, box limits denote the upper and lower quartiles, and whiskers denote the 1.5× interquartile range. **B** H&E image and manual annotation from the original study. **C** Clustering results by nonspatial and spatial methods, Seurat, Giotto, SpaGCN, BayesSpace, SpaceFlow, conST, STAGATE, and GraphST on slice 151673 of the DLPFC dataset. Manual annotations and clustering results of the other DLPFC slices are shown in Supplementary Fig. S1. **D** Laminar organization of the

mouse olfactory bulb annotated using the DAPI-stained image. **E** Spatial domains identified by Seurat, STAGATE, and GraphST in the mouse olfactory bulb Stereo-seq data. **F** Visualization of the spatial domains identified by GraphST and the corresponding marker gene expressions. The identified domains are aligned with the annotated laminar organization of the mouse olfactory bulb. **G** Allen Mouse Brain Atlas with the hippocampus region annotated. **H** Spatial domains identified by Seurat, STAGATE, and GraphST in mouse hippocampus tissue acquired with Slide-seqV2. **I** Visualization of the spatial domains identified by GraphST and the corresponding marker gene expressions. The identified domains are aligned with the annotated hippocampus region of the Allen Mouse Brain Atlas.

GraphST was able to demarcate the MH and LH regions with high concordance based on their respective markers, *Nwd2* and *Gpr151*. For the third ventricle, the *Enpp2* expression did not align well with GraphST's detected V3 region, but the latter better resembled the V3 region in the annotated brain reference. For comparison, Seurat and STAGATE's V3 regions were closer to the shape of *Enpp2* expression region but did not match the anatomical shape well.

### GraphST's spatial clustering of mouse embryo Stereo-seq data revealed finer-grained tissue structures

In this final clustering comparison, we used Stereo-seq acquired datasets of mouse embryos at E9.5 and E14.5, with 5913 bins and 25,568 genes, and 92,928 bins and 18,566 genes, respectively[22]. Tissue domain annotations were obtained from the original study wherein Leiden clustering from SCANPY[37] was applied to the union of neighborhood graphs constructed using spatial proximity and transcriptomic similarity, and the computed clusters were annotated based on differentially expressed genes. We first examine the clustering results of the E9.5 embryo. Although the original annotation had 12 reference clusters (Fig. 3A), we set the number of clusters in our testing to 22 to acquire a higher resolution of tissue segmentation. STAGATE output was notable, with many identified clusters collectively forming a thick layer around the embryo (Fig. 3B). The identified clusters also did not match the annotation well. In contrast, GraphST's clusters better matched the annotated regions (Fig. 3B and Supplementary Fig. S3A–B). More importantly, they showed high concordance with known marker genes of the major organs. Particularly, the liver region marked by *Afp*, *Fgb*, *Alb*, *Itih2*, dermomyotome by *Myog*, mesenchyme by *Meox1*, Meox2, *Pcp4*, head mesenchyme by *Crym*, *Six2*, *Alx4*, Sclerotome by *Pax1*, *Pax9*, *Meox1*, *Meox2*, heart by *Myl2*, *Myl7*, *Nppa*, *Myh6*, *Myh7*, *Tnni3*, *Ttn*, and connective tissue by *Postn*, were all well matched by the clusters of GraphST (Fig. 3C and Supplementary Fig. S3C–D). GraphST also demarcated two clusters for the embryo heart's atrium and ventricle chambers, with compartment-specific markers such as *Myl7* and *Nppa* marking the atrium and ventricle chambers, respectively. *Nppa* is unique with a spatiotemporal expression pattern during heart development, where it is first expressed in the developing mouse heart at E8.0–8.5[38], specifically at the ventral portion of the linear heart tube, which develops into the ventricles. At E9.0–9.5, *Nppa* is expressed mostly in the left and right ventricles within the looped embryonic heart, but its expression becomes restricted to the atria upon birth. We also examined the transcription factors essential for heart development and cardiac septation, namely *Gata4*, *Tbx5*, and *Gata6*, and their regulon activities showed spatial distributions that matched GraphST's clusters (Fig. 3C and Supplementary Fig. S3D). Although STAGATE was also able to identify the two halves of the heart, albeit with a lower concordance with cardiac gene expression, it was unable to identify the other major organs.

We next compared the clustering results with the E14.5 mouse embryo (Fig. 3D). Here, we set the number of clusters to 16, matching the original annotation. STAGATE produced overly smoothened clusters and failed to reveal any fine-grained tissue complexity (Fig. 3E). In contrast, GraphST's clusters were able to capture much of the fine-grained structures in the embryo, and they

were also highly concordant with the original annotation, accurately identifying major areas such as the heart, liver, and muscle regions (Fig. 3F and Supplementary Fig. S4C). Furthermore, GraphST was able to accurately outline the epidermis as confirmed by the expressions of *Krt5*, *Krt14*, and *Krt15* (Fig. 3G), while the original annotation assigned parts of the epidermis to the cavity instead. GraphST was also better at delineating the meninges region as a continuous closed loop. In contrast, the original annotation's meninges cluster appeared more fragmented and discontinuous. Notably, the original annotation demarcated the choroid plexus as a separate region, while GraphST clustered it as part of the meninges. As the choroid plexus resides in the meninges' innermost layer (the pia mater) of three, it is difficult to be detected as a separate region. Within the cranial region, GraphST demarcated several subareas that were not in the original annotation. One key identified area is the cerebral cortex, which we verified with the expression of cortical glutamatergic neuroblast markers *Neurod6*, *Tbr1*, and *Eomes* (Fig. 3G). Though GraphST did not identify the olfactory epithelium found in the original annotation, it was able to delineate a separate olfactory epithelium cluster when we increased the number of clusters to 20 (Supplementary Fig. S4D). Finally, GraphST also delineated a cluster of osteoblasts marked by *Nov* and *Mfap5*. To characterize this cluster, we employed gene set enrichment analysis with Enrichr[39], which suggested that this cluster is associated with mouse osteoblasts from day 14 and day 21 of development (Supplementary Fig. S4E). Overall, the mouse embryo clustering results affirmed GraphST's strength over STAGATE in identifying anatomically distinct regions in complex tissue samples.

### GraphST corrects batch effects for vertical and horizontal integration of multiple tissue sections

Serial tissue slices are experimentally probed to increase accuracy and enable reconstruction on the vertical axis. An obstacle to this data integration is the presence of batch effects. To date, there are few existing methods developed to integrate multiple tissue sections for joint analysis. Here we compared GraphST with nonspatial methods, Harmony and scVI, and the spatial method STAGATE in vertically integrating serial tissue sections, using two mouse breast cancer serial sections acquired with 10x Visium as the test data. We first assessed the integration results visually with UMAP. In the uncorrected data, strong batch effects were clearly present (Fig. 4A). After integration, both scVI and Harmony were able to mix the two slices, but some batch-specific separation was still visible postintegration. GraphST was able to evenly mix both slices while STAGATE was only able to pull the slices closer (Fig. 4B). For quantitative assessment, we employed the widely used metric iLISI (integration local inverse Simpson's Index)[16]. GraphST achieved the highest score of 1.846 while Harmony, scVI, and STAGATE scored lower with 1.566, 1.568, and 1.233, respectively (Fig. 4C). In the subsequent clustering, GraphST's clusters in both slices were also highly overlapping, unlike the clusters from Harmony, scVI, and STAGATE. This showed that GraphST was able to correct the batch effects and accurately align the regions across different serial slices. The clusters identified by Harmony, scVI, and STAGATE were also more fragmented than GraphST's.

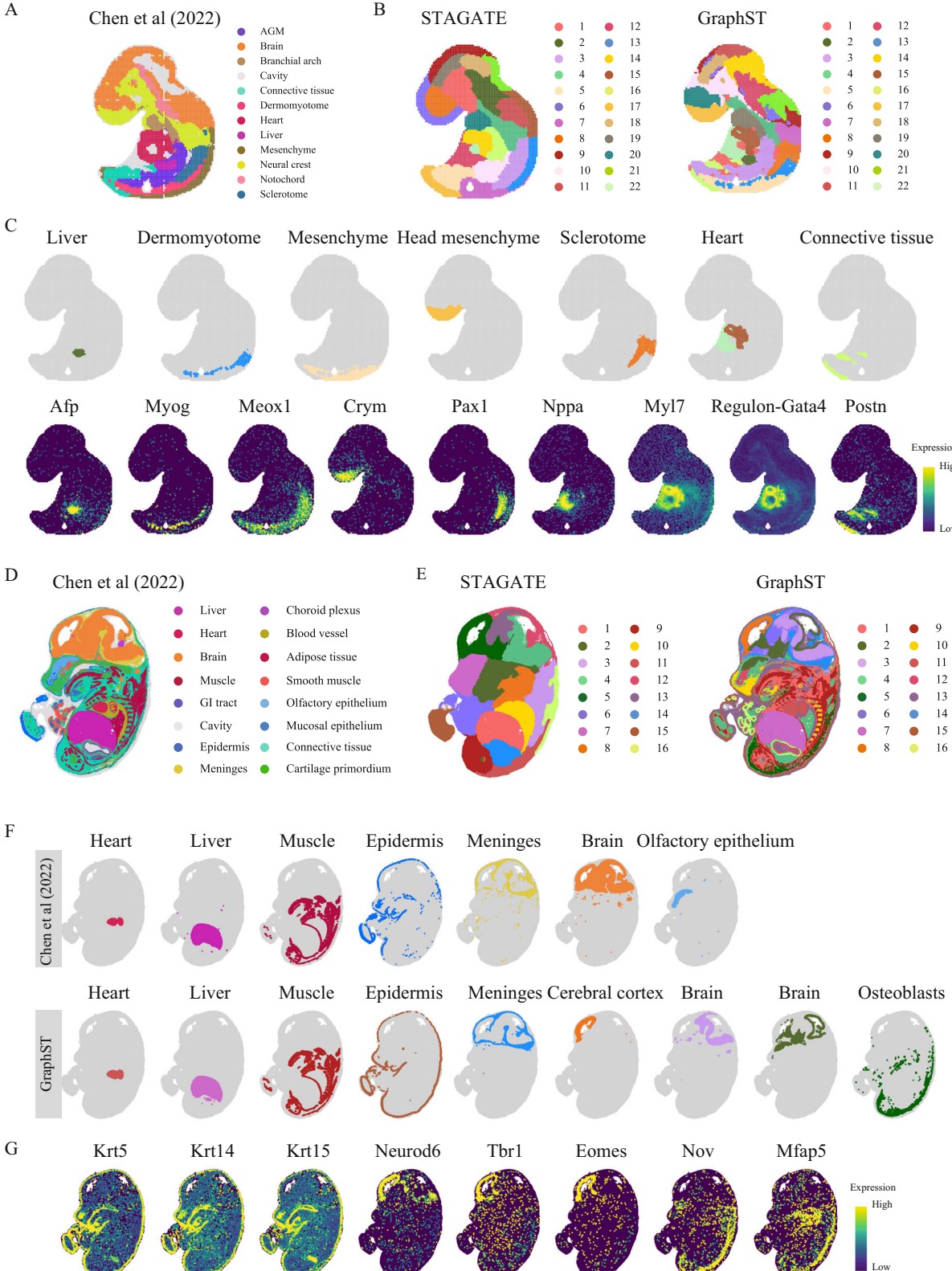

**Fig. 3 | GraphST enables accurate identification of different organs in the Stereo-seq mouse embryo. A** Tissue domain annotations of the E9.5 mouse embryo data taken from the original Stereo-seq study wherein the clusters were first identified using Leiden clustering from SCANPY and then annotated using differentially expressed genes. **B** Clustering results of STAGATE and GraphST on the E9.5 mouse embryo data. **C** Visualization of selected spatial domains identified by GraphST and the corresponding marker gene expressions. **D** Tissue domain annotations of the E14.5 mouse embryo data obtained from the original Stereo-seq study. **E** Clustering results by STAGATE and GraphST on the E14.5 mouse embryo. **F** Visualization of selected spatial domains identified by the original Stereo-seq study and GraphST, respectively. **G** Visualization of marker gene expressions supporting the identified domains.

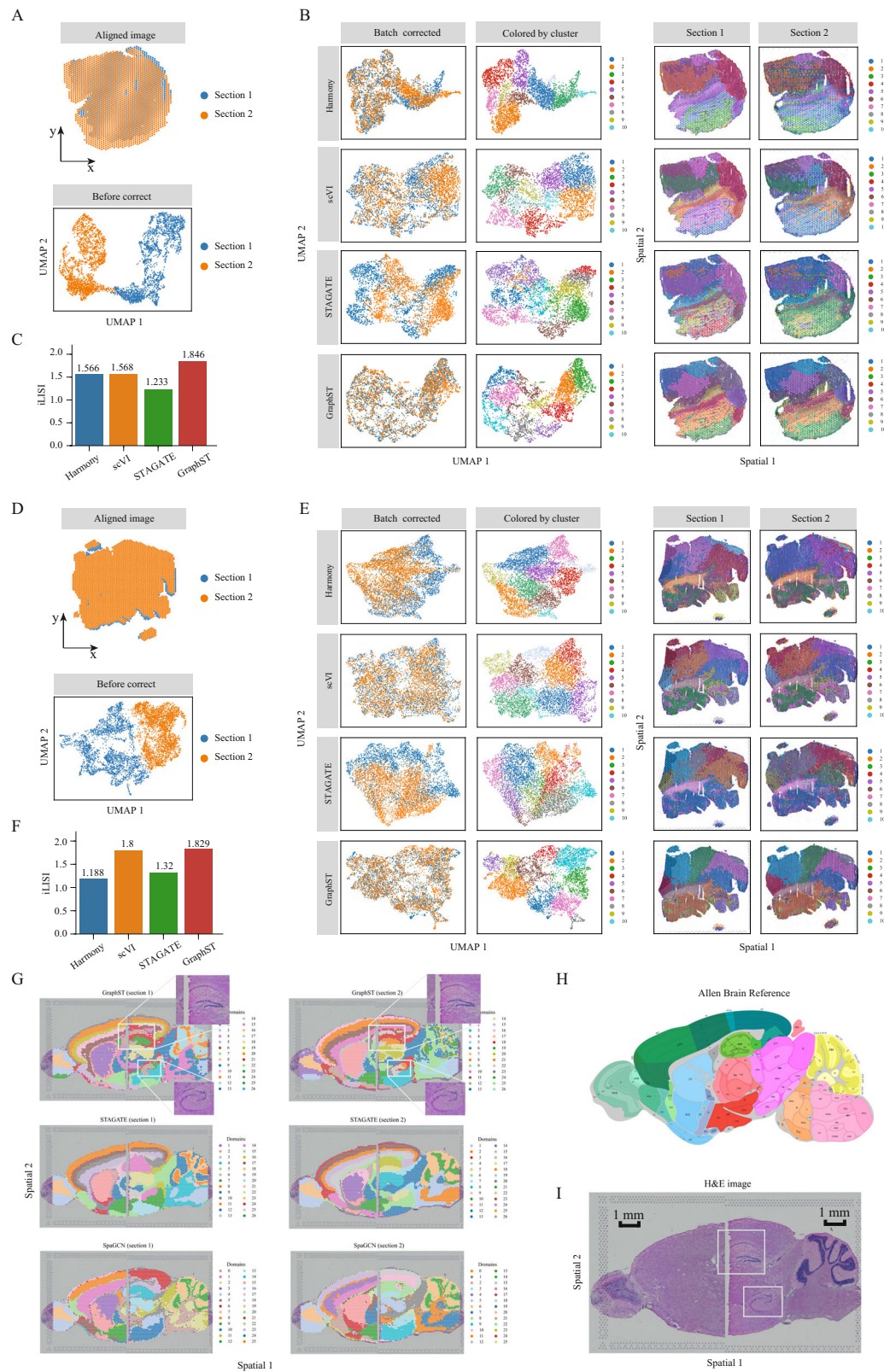

We next tested the methods on another mouse breast cancer dataset with batch effects present (Fig. 4D). On the UMAP plots, batch differences remained visible in the Harmony and STAGATE outputs (Fig. 4E). scVI removed the batch effects much better than Harmony and STAGATE, while GraphST performed the best by evenly mixing the two slices. In terms of iLISI, Harmony and STAGATE significantly underperformed GraphST, while scVI was comparable to GraphST

(Fig. 4F). In the subsequent clustering, GraphST again found clusters that aligned well across the two slices. Most of Harmony's clusters did not align across the two slices. While scVI generated clusters that were more consistent than Harmony's, there were still fragmented clusters, especially in section 2. STAGATE's clusters showed better overlap than in the previous example but were still poorer than GraphST's, with significant differences visible in clusters such as 2, 3, 4, and 8.

**Fig. 4 | GraphST enables accurate vertical and horizontal integrations of ST data on mouse breast cancer and mouse brain anterior and posterior data, respectively. A** First set of mouse breast cancer sample images aligned with the PASTE algorithm and plotted before batch effect correction. **B** UMAP plots after batch effect correction and spatial clustering results from Harmony, scVI, STA-GATE, and GraphST. Spots in the second column are colored according to the spatial domains identified by the respective clustering methods. **C** Barplots of iLISI metric for batch correction results from different methods on the first set of samples. **D** Second set of mouse breast cancer sample images aligned and UMAP plot before batch effect correction. **E** UMAP plots after batch effect correction and the spatial clusters detected by Harmony, scVI, STAGATE, and GraphST on sections 1 and 2, respectively. Similarly, spots in the second column are colored according to the spatial domains identified by the respective methods. **F** Barplots of iLISI metric for batch correction results from different methods on the second set of samples. **G** Horizontal integration results with two mouse brain samples, of which each consists of anterior and posterior brain sections. Top: spatial joint domains identified by GraphST on sections 1 and 2. Middle: spatial joint domains identified by STAGATE. Bottom: spatial joint domains identified by SpaGCN. **H** Annotated brain section image from Allen Mouse Brain Atlas for reference. **I** H&E image of mouse brain anterior and posterior.

As tissue samples can be significantly larger than the capture slides used for spatial transcriptomics, horizontal integration enables the data from multiple capture slides to be stitched together. Here we tested GraphST, STAGATE, and SpaGCN's horizontal integration capabilities using two sections of the mouse brain 10x Visium data with both divided into two slices, anterior and posterior (Fig. 4G). We first aligned the tissue slices of each section and performed clustering. For comparison purposes, we set the number of clusters to 26 for all three algorithms, the same number used in the demonstration of SpaGCN[9]. Using the mouse brain atlas annotation[40] (Fig. 4H), we compared the results for both sections. We find that most of the clusters from STA-GATE, SpaGCN, and GraphST showed good agreement with known anatomy. However, many of the SpaGCN clusters were fragmented and were not aligned along the shared edge of the two tissue sections. GraphST and STAGATE performed better than SpaGCN and showed much better alignment along the edge of the anterior and posterior sections. The clusters from both GraphST and STAGATE were accurate at capturing the structures of the cerebral cortex layers, cerebellum, and corpus callosum. However, some key regions in the tissue were not captured by STAGATE. For example, STAGATE failed to identify the dorsal (top) and the ventral (bottom) horn of the hippocampus regions, as indicated with white boxes on the H&E images (Fig. 4I). In contrast, GraphST was able to reveal these regions. This example again highlights GraphST's consistent capability in aligning shared domains across tissue sections while removing batch effects.

## GraphST projects scRNA-seq to ST for cell-type deconvolution of ST

Using scRNA-seq data as reference, GraphST can perform cell-type deconvolution of ST data by mapping the single-cell gene expression profiles onto the ST spots. Upon mapping, GraphST produces a cell-to-spot mapping probability matrix that can be combined with cell-type annotation to derive the spatial probability distribution of each cell type, which is equivalent to cell-type deconvolution of spatial spots. We compared GraphST's deconvolution with cell2location, which, to our knowledge, is the top-performing method for ST deconvolution[41]. We first used the simulated datasets derived from seqFISH+ and STARmap acquired ST data used in the benchmarking study. The original high-resolution data was overlaid with a coarse grid, and cellular gene expression was summed within each grid square to simulate spots capturing multiple cells. We calculated four indices, Pearson correlation coefficient (PCC), structural similarity index measure (SSIM), root mean squared error (RMSE), and Jensen–Shannon divergence (JSD) to compare performance (Fig. 5A). With the seqFISH+ derived simulated dataset, GraphST slightly outperformed cell2location in the first three indices PCC and SSIM (higher is better), and JSD (lower is better), while being slightly poorer in terms of RSME (lower is better). For the STARmap-derived dataset, GraphST also achieved better PCC, SSIM, and JSD scores while having similar average performance in terms of RSME.

To demonstrate the ability of GraphST at integrating scRNA-seq data with ST data wherein each spot contains more than one cell, we first tested GraphST and competing methods with a publicly available

10x Visium dataset acquired from human lymph node tissue containing germinal centers (GC)[42]. The reference scRNA-seq data used for deconvolution was created from three studies on human secondary lymphoid organs and is composed of 34 cell types and 73,260 cells[43–45]. The human lymph node is a dynamic environment with different cell types spatially intermixed, creating a challenge for cell-type deconvolution. We first visually compared the results of GraphST and cell2location by examining the mapping of cell types onto GC and non-GC locations. The ground-truth annotations of the germinal centers used here were obtained from the original cell2location study. Comparatively, GraphST mapped more GC preplasmablast (prePB) and cycling, dark zone (DZ), light zone (LZ) B cells to the annotated germinal center locations (Fig. 5B and C). Furthermore, the cycling and DZ B cells were correctly colocalized. The LZ B cells and prePB cells also spatially coincided as expected. The naïve B and preGC B cells were also correctly mapped to regions surrounding the GCs. We also quantified performance using ROC curves for the light zone, dark zone, and preplasmablast cells (Fig. 5D). In all three cases, GraphST achieved the best performance with cell2location as a close second, while the remaining methods were significantly poorer. Finally, we quantified the GC cell-type mapping by computing the odds ratio between the sum of deconvolution scores of GC spots and non-GC spots. For all computed cell types (LZ, DZ, and prePB), GraphST's results again scored higher than cell2location, indicating a higher concordance between ground-truth annotation and GraphST's deconvolution (Fig. 5E). The mapping results of all 34 cell types are found in the Supplementary Figs. S5 and S6 for GraphST and cell2location, respectively.

We also tested GraphST and cell2location on the human dorsolateral prefrontal cortex (DLPFC) dataset. The DLPFC has a clear layered tissue structure (Fig. 5F) which can help better distinguish performance differences between the methods. We first selected slice #151673 (3639 spots and 33,538 genes) and paired it with the scRNA-seq data used in the CellDART[46] study with 78,886 cells and 30,062 genes and annotated with 33 cell types. The cell types spatially mapped by GraphST exhibited cortical layer-specific distribution patterns (Fig. 5F). Here, we see Ex_10_L2_4, Ex_7_L4_6, Ex_1_L5_6, Ex_8_L5_6, and Ex_4_L_6 form discernable ordered layers from the outermost to the innermost, demonstrating GraphST's capability to distinguish the progressive layering of the cortical regions that are representative of the actual physiological anatomy. GraphST also mapped the oligos cells onto the WM layer, which is known to be enriched with oligodendrocytes. In cell2location's output, the oligos cell mappings were similar to GraphST's but without clear edges. Meanwhile, the other cortex cell mappings were much more scattered and did not correspond to their expected positions in the layers. The mapping results of all 33 cell types are given in Supplementary Figs. S7 and S8 for GraphST and cell2location, respectively.

We further tested GraphST and cell2location for cell-type deconvolution with the mouse brain anterior dataset. GraphST similarly showed mappings with higher density and sharper edges, while cell2location's mappings tend to be more diffuse. The full analysis results are presented in Supplementary File 1.

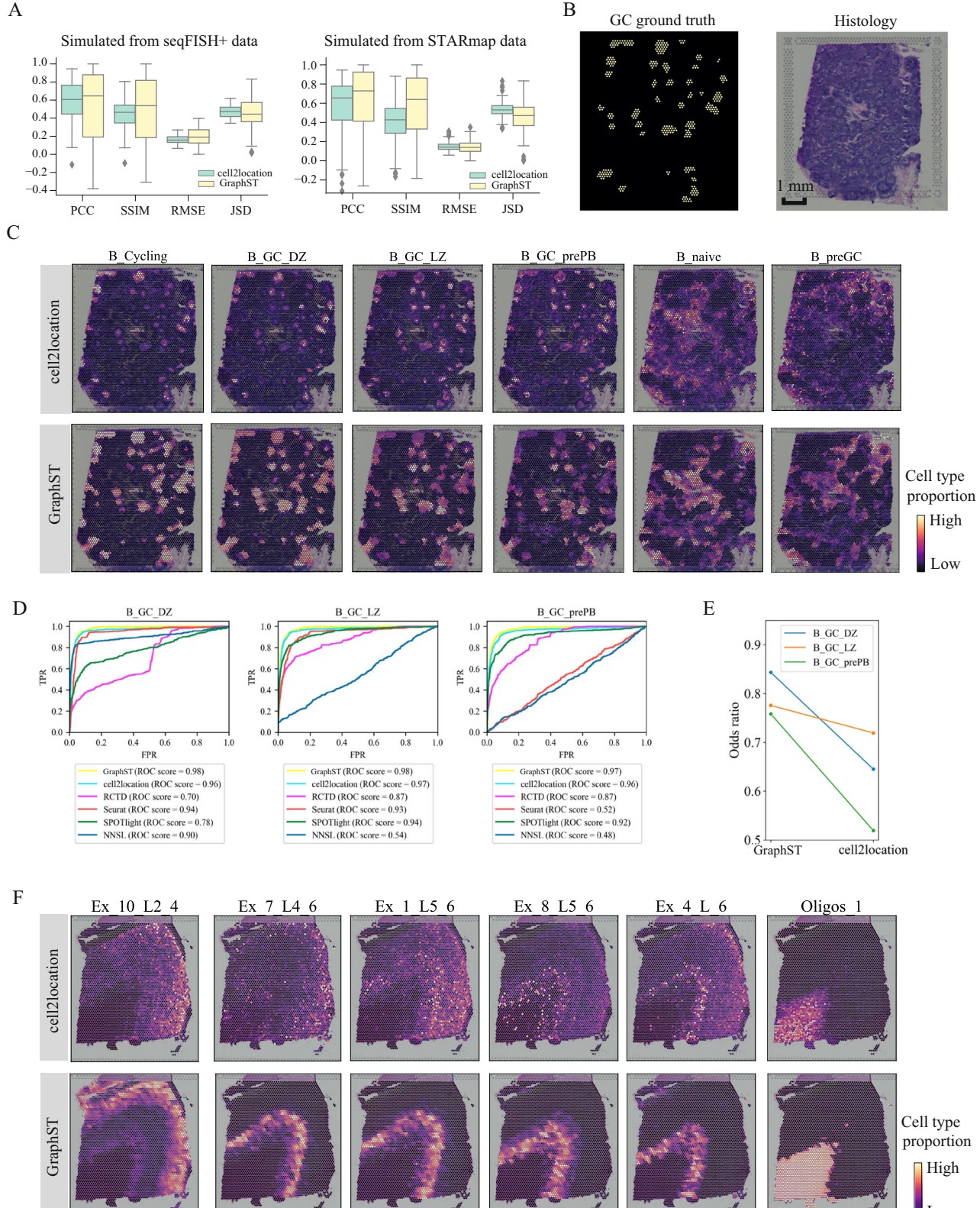

## Accurate spatial mapping of cells to human breast cancer 10x Visium data revealed T-cell suppression in IDC regions

In this comparison, we first performed clustering on a human breast cancer 10x Visium dataset with 3798 spots and 36,601 genes. The data were manually annotated by a pathologist based on the H&E image and the spatial expression of reported breast cancer marker genes (Supplementary Figs. S12A and S13). The data was annotated with 20

regions, so we specified the clustering parameters to obtain the same number of clusters (Supplementary Fig. S12B). For Seurat, Giotto, conST, and STAGATE, many of the computed clusters were fragmented and discontinuous, while SpaceFlow, BayesSpace, SpaGCN, and GraphST produced less disjoint clusters. This result was also reflected in the ARI scores where Seurat, Giotto, SpaceFlow, and STAGATE had lower scores in the range of 0.39–0.47, while

**Fig. 5 | Comparing the accuracy of GraphST with top deconvolution method cell2location in predicting spatial distributions of scRNA-seq data with simulated data, human lymph node, and the slice 151673 of DLPFC. A** Boxplots of PCC, SSIM, RMSE, and JSD metrics for cell2location and GraphST results on simulated data created from seqFISH+ and STARmap experimental data. In the boxplot, the center line denotes the median, box limits denote the upper and lower quartiles, and whiskers denote the 1.5× interquartile range. *n* = 8 (12) different predicted cell types for simulated data created from seqFISH+ (STARmap) experimental data. **B** Left, annotations of germinal center (GC) locations from cell2location's study (GC locations annotated with yellow). Right, H&E image of human lymph node data.

**C** Comparison between cell2location and GraphST on the spatial distributions of selected cell types, namely B_Cycling, B_GC_DZ, B_GC_LZ, B_GC_prePB, B_naive, and B_preGC. **D** Quantitative evaluation via AUC of three cell types (B_GC_DZ, B_GC_LZ, and B_GC_prePB) localized in the GCs using the annotated locations shown in (**B**). **E** Quantitative evaluation of GC cell type mapping of three cell types (B_GC_DZ, B_GC_LZ, and B_GC_prePB) between cell2location and GraphST using the odds ratio metric. **F** Comparison between cell2location and GraphST on the spatial distribution of cell types Ex_10_L2_4, Ex_7_L4_6, Ex_1_L5_6, Ex_8_L5_6, Ex_4_L_6, and Oligos_1 with slice 151673 of the DLPFC dataset.

BayesSpace, SpaGCN, and GraphST had higher scores between 0.54 and 0.57. We next applied GraphST to deconvolute the cell types present in each spot by mapping cells from the DISCO scRNA-Seq human breast tissue atlas[47] onto the breast cancer ST dataset (Fig. 6A). The fibroblasts, perivascular cells, lymphatic endothelial, and vascular endothelial cells were expectedly mapped onto the Healthy and Tumor edge regions, while the myoepithelial cells were mostly mapped to the Tumor edge regions. The luminal and luminal progenitor cells were primarily mapped onto the IDC and DCIS/LCIS regions.

Focusing on the immune cells, we found a significant presence of diverse immune subsets, namely T cells and macrophage/DC/monocytes (myeloid cells) in the IDC regions (IDC 5, 6, 7, and 8). The correspondence of the single cells with their mapped regions was also visualized with UMAP (Fig. 6B, C). Of the DCIS/LCIS regions, 4 and 5 only had macrophage/DC/monocytes present, while 1 and 2 had no immune cells mapped onto them. The cell-type deconvolution found only small amounts of B and plasma cells in the sample. B cells were mainly mapped to IDC 3 and a portion of Tumor edge 3, while small concentrations of plasma cells were found in the various Tumor edge regions. Macrophages being present in tumor tissues is of clinical concern as they are linked to tumor progression and hence poor patient survival[48]. The lack of B cells is also significant as it is associated with poor patient prognosis[49].

As the T cells were found across different tissue regions (DCIS/LCIS, Healthy, IDC, and Tumor edge), we further investigated their differences (Fig. 6C). We first computed the differentially expressed genes (DEGs), followed by the associated differentially regulated pathways. We found the T cells in IDC regions to have the largest number of DEGs, including genes associated with T-cell exhaustion (LAG3, CTLA4, TIGIT, HAVCR2, PDCD1), and CXCL13 (Fig. 6D). In the pathway analysis, the most significant differentially regulated pathways were associated with leukocyte activation and response (Fig. 6E). These results highlighted T-cell dysregulation within the tumor environment. Such T cells were also observed by Zhang et al.[50] and Bassez et al.[51] in other human breast cancer samples, and both studies demonstrated their reactivation with anti-PD-L1 therapy to harness their cytotoxicity against the tumor cells.

When comparing the spatial mapping of cells from adjacent normal and solid tumor sites, adjacent normal cells were mainly mapped onto the Healthy (1, 2) and Tumor edge (1, 2, 4, 5) regions (Fig. 6F, Supplementary Fig. S12C). The DCIS/LCIS regions (1, 2) also showed high levels of adjacent normal cell mapping. Conversely, the solid tumor-derived cells mostly mapped to the IDC regions. Unlike other DCIS/LCIS regions, DCIS/LCIS 4 also had more solid tumor-derived cells mapped to it. Other regions, notably Tumor edge 3 and DCIS/LCIS 5, had a mix of cell mappings from both adjacent normal and solid tumor cells. These mappings suggested that DCIS/LCIS 4 and 5 were more advanced in tumor development than other DCIS/LCIS regions. For Tumor edge 3, its lower region was mapped with tumor cells, while the upper region was primarily mapped with adjacent normal cells. Interestingly, GraphST merged the upper portion of Tumor edge 3 with DCIS/ICIS 1, which was also mapped with adjacent normal cells (Supplementary Fig. S12B and C). Another notable feature of Tumor edge 3's lower region was that it had one of the two rare

concentrations of B cells in this tissue sample. Further investigation is needed to reveal the reasons behind B cell infiltration into this region but not others, as B cell infiltration is considered positive for patient prognosis[49]. In conclusion, this case study demonstrated the value of cell type and sample type deconvolution in exploring the heterogeneity present within tissues that may otherwise not be apparent from the histological analysis. Moreover, such analysis can also help to determine the appropriate number of regions for clustering purposes.

## Discussion

Spatial transcriptomics is a powerful experimental approach that measures gene expression while retaining the associated spatial context. This combination of gene expression and spatial information enables spatially informed clustering to identify coherent tissue domains that are biologically relevant. While powerful, ST faces technological limitations that computational approaches can help mitigate. First, ST capture areas are currently limited in size, necessitating multiple adjacent tissue slides to capture larger tissue samples. Horizontal data integration is thus needed for integrated studies. Extension to the three-dimensional characterization of a sample will require vertical data integration of serial tissue slides. Second, the two major experimental approaches in ST have their respective limitations, namely sequencing-based approaches with a low spatial resolution with each spot containing more than one cell and imaging methods with lower gene coverage at a higher subcellular resolution. Here, additional RNA-seq data offers the opportunity to infer the cellular composition or cellular transcriptome for the two approaches, respectively. In particular, scRNA-seq with single-cell resolution genome-wide gene coverage but no spatial information is highly relevant. We can perform ST spot cellular deconvolution with scRNA-seq, as well as assign cells and their transcriptomes to spots. In this paper, we presented GraphST, a graph self-supervised contrastive learning method that can effectively and efficiently perform spatial clustering, spatial transcriptomics data integration, and scRNA-seq data transfer onto ST data. GraphST employs graph neural networks to learn informative representations of the gene expression profiles with spatial locations. The learned representations are further refined using graph self-supervised contrastive learning with the spatial context information to be more informative and discriminative. For scRNA-seq data transfer, we use an auto-encoder to first extract features from the scRNA-seq data while reducing the noise before using spatially informed contrastive learning to train the cell-to-spot mapping matrix.

When clustering spots in diverse ST datasets, GraphST was better than competing methods at identifying biologically accurate structures in highly heterogeneous tissue samples. We also demonstrated GraphST's ability at vertical and horizontal data integration with multiple tissue samples. GraphST detected domains that were more continuous across samples and more accurate when integrating ST data horizontally. When integrating ST data vertically, it identified biologically coherent spatial domains that aligned across samples while removing the batch effect present. Finally, we validated GraphST's ability to map scRNA-seq data onto ST-acquired spots to quantify the

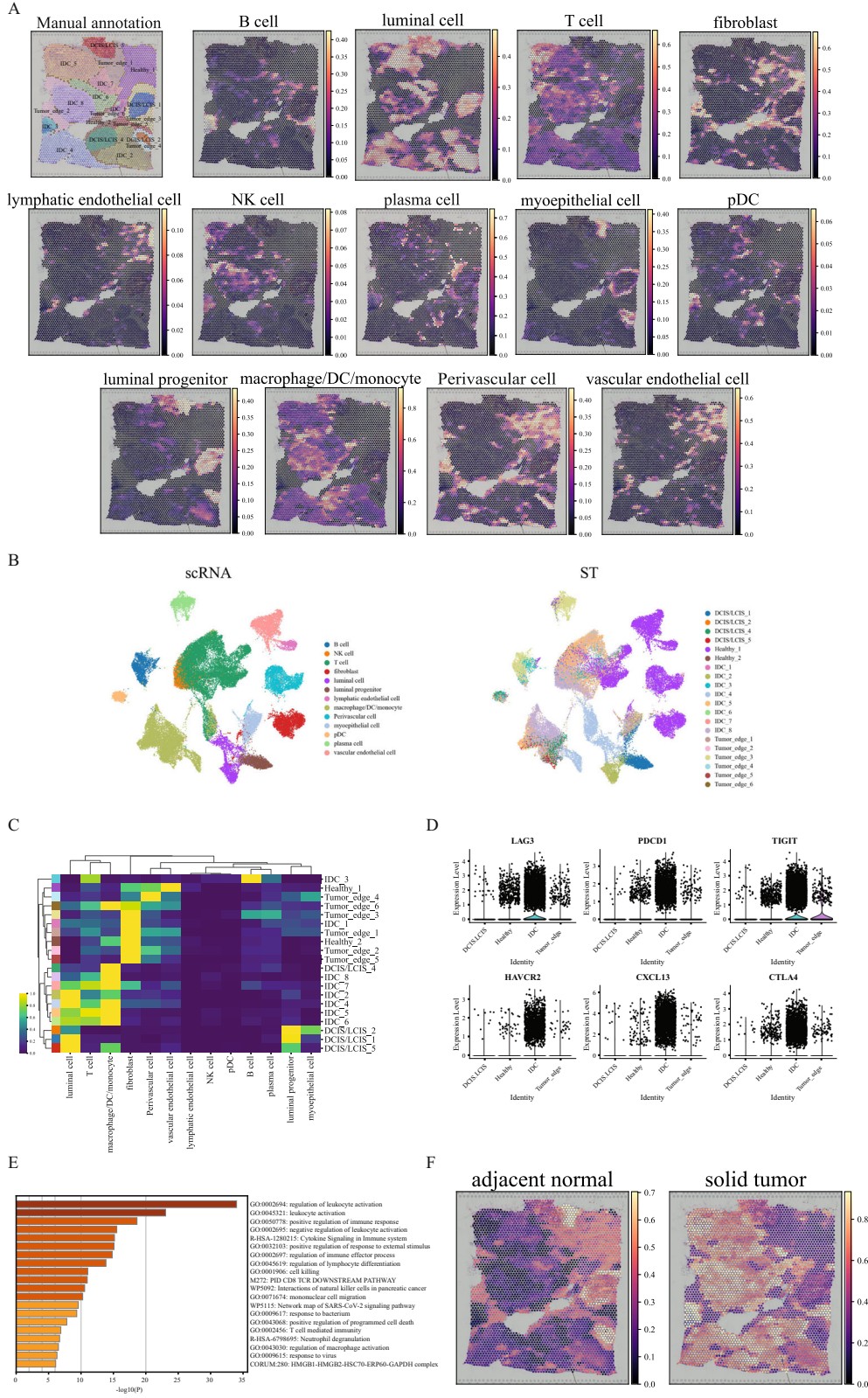

**Fig. 6 | GraphST enables comprehensive and accurate spatial mapping of scRNA-seq data in human breast cancer data. A** Manual annotation and spatial distribution of major cell types mapped by GraphST, namely B cell, luminal cell, T cell, fibroblast, lymphatic endothelial cell, NK cell, plasma cell, myoepithelial cell, pDC, luminal progenitor, macrophage/DC/monocyte, Perivascular cell, and vascular endothelial cell. **B** Visualization of scRNA-seq data and spatial localization of cell types with UMAP generated from the output cell representations of GraphST.

**C** Heatmap of the spatial distribution of cell types. **D** The gene expression of six T-cell exhaustion-related markers in different annotated domains. **E** Functional enrichment results of the IDC domain specific differentially expressed genes. Statistical significance was assessed by the hypergeometric test, and p-values were adjusted by the Benjamini–Hochberg p-value correction algorithm. The statistical test was one-sided. **F** Predicted spatial distribution of cells from two sample types, adjacent normal and solid tumor.

spatial distribution of cell types. We exemplified GraphST's utility in analyzing tissue microenvironment by deconvoluting both the cancer cells and immune milieu present in breast cancer tissue.

The key features of GraphST responsible for its superior performance are the use of graph neural networks and contrastive learning in capturing gene expression together with spatial information. While existing methods such as SpaGCN and STAGATE also use graph neural networks to learn gene expression and spatial information, they show poorer clustering performance and lack the ability to accomplish batch integration. The main differences between GraphST and these methods lie in the use of graph self-supervised contrastive learning to strengthen the latent representation learning and additional preservation of local spatial context information of spots. This makes the learned representation more informative and discriminative, thus improving clustering performance. Moreover, while there are similarities between GraphST and STAGATE in the integration of multiple samples, STAGATE's limited capability in batch effect removal greatly limited its performance in vertical integration. On the other hand, although conST and SpaceFlow also adopted graph contrastive learnings for spatial clustering, GraphST possesses major technical differences and performance advantages compared to the other two methods. GraphST is different from conST and SpaceFlow in its definition of positive/negative pairs, formulation of the objective function and contrastive loss, and training procedure (details in Supplementary File 1). These differences enable GraphST to outperform conST and SpaceFlow in the spatial clustering task. We have conducted several ablation studies to confirm that each of these differences improves the effective integration of gene expression and spatial context to obtain informative and discriminative spot representations (Supplementary File 1). Moreover, both conST and SpaceFlow were mainly developed for spatial clustering only. In addition to spatial clustering, GraphST can also be applied to two other important ST data analysis tasks, multisample integration and cell-type deconvolution of ST.

In cell-type deconvolution, GraphST also offers improvements over competing methods. Most existing deconvolution methods, such as SPOTLight[28] and cell2location[29], require accompanied cell-type annotation. Moreover, they do not consider spatial information in their deconvolution. In contrast, GraphST employs augmentation-free contrastive learning and spatial information to learn the cell-to-spot mapping matrix. The learning process does not require prior cell-type information, and thus the mapping matrix is able to flexibly project arbitrary cell attributes (e.g., cell type and sample type) onto the spatial space. Thus, GraphST can directly project scRNA-seq data onto the spatial spots to achieve cell-level deconvolution. While the mapping matrix learning process is similar between GraphST and Tangram, there are notable differences. Tangram only uses the raw gene expression, while GraphST employs augmentation-free contrastive learning to learn the mapping matrix and take in both the spatial location information and the informative and noise-reduced gene expression from the graph deep learning modules as input.

We designed GraphST to be user-friendly and capable of processing data acquired from different experimental platforms. GraphST has been validated on 10x Visium, Slide-seqV2, and Stereo-seq data, and we plan to extend it to other ST platforms, such as MERFISH and Nanostring CosMx SMI. GraphST is also designed to be computationally efficient in handling challenges from large datasets. We anticipate that future developments in ST technologies will bring about subcellular resolution with full gene expression profiling, as well as general growth in dataset sizes. The largest dataset we tested contained about 100,000 spots (E14.5 mouse embryo), and it required 30 mins of wall-clock time on a server with Intel Core i7-8665U CPU and NVIDIA RTX A6000 GPU. We believe that our algorithm can handle existing datasets as well as newer ones in the near future. We also plan to further improve performance through distributed computing and batch training implementations.

## Methods

### Data preprocessing
For spatial clustering, GraphST takes in gene expression counts and spatial position information. Raw gene expression counts are first log-transformed and normalized by library size via the SCANPY package[37]. The normalized gene expression counts are then scaled to unit variance and zero mean. Finally, the top 3000 highly variable genes (HVGs) are selected to be input into the GraphST model. For ST cell-type composition deconvolution, scRNA-seq data is similarly preprocessed, where the raw gene expression counts are first log-transformed and normalized by library size and then scaled to unit variance and zero mean. Subsequently, the top 3000 highly variable genes are selected. To ensure consistent feature information, the common preprocessed HVGs of the scRNA-seq and ST data are used as input to GraphST to learn the latent representations of cells and spots, respectively.

### Graph construction for spatial transcriptomics data
The strength of spatial transcriptomics is the associated spatial information that can be exploited to identify similar cell states that are also spatially co-located and thus demarcate tissue substructures. To make full use of the spatial information, we convert it into an undirected neighborhood graph $G = (V, E)$ with a predefined neighbor number $k$. In the graph $G$, $V$ represents the set of spots while $E$ is the set of connected edges between spots. $A \in \mathbb{R}^{N_{spot} \times N_{spot}}$ is defined as the adjacency matrix of graph $G$ with $N_{spot}$ denoting the number of spots. If spot $j \in V$ is the neighbor of spot $i \in V$, $a_{ij} = 1$, otherwise 0. Thus, for a given spot, its neighbors are defined by its proximity to other spots based on the Euclidean distance computed from the spatial location information. Finally, we select $k$ spots as its neighbors from the top nearest neighbors. From our testing, GraphST achieved its best performance in most of the tested datasets with $k$ set to 3.

### Graph self-supervised contrastive learning
For spatial clustering, a graph self-supervised contrastive learning framework is proposed to learn the spot representations from the input gene expression profiles and spatial information. Figure 1A illustrates the overview of this framework, divided into three major steps: (1) data augmentation, (2) GNN-based encoder for representation learning, and (3) self-supervised contrastive learning for representation refinement. GraphST takes the gene expression profiles and the neighborhood graph as input and outputs the spot representations for spatial clustering and multiple ST data integration. The details of each step are described next.

### Data augmentation
For subsequent contrastive learning, we first generate a corrupted neighborhood graph via data augmentation. Specifically, given a neighborhood graph $G$ and the normalized gene expression matrix $X$, we create the corrupted graph by randomly shuffling the gene expression vectors among the spots while keeping the original graph's topological structure unchanged. Let $G' = (V', E')$ and $X'$ denote the corrupted graph and shuffled gene expression profiles, respectively.

### GNN-based encoder for representation learning
We designed a GNN-based encoder to learn spot representations that capture the informative parts of the gene expression profiles and spatial locations. The encoder takes the neighborhood graph $G$ and the normalized gene expression profiles in $X$ as inputs, and the decoder outputs the reconstructed gene expressions $H_s$. Specifically, we utilize a graph convolutional network (GCN)[52] as encoder to learn a latent representation $z_i$ for spot $i$ by iteratively aggregating the representations of its neighbors. Formally, the $l$-th layer representations in the

encoder can be formulated as follows,

$$Z_s^l = \sigma\left(\widetilde{A} Z_s^{l-1} W_e^{l-1} + b_e^{l-1}\right), \tag{1}$$

where $\widetilde{A} = D^{-\frac{1}{2}} A D^{-\frac{1}{2}}$ represents the normalized adjacent matrix where $D$ is a diagonal matrix with diagonal elements being $D_{ii} = \sum_{j=1}^{N_{spot}} A_{ij}$. $W_e$ and $b_e$ denote a trainable weight matrix and a bias vector, respectively. $\sigma(\cdot)$ is a nonlinear activation function such as ReLU (Rectified Linear Unit). $Z_s^l$ denotes the $l$-th layer output representation and $Z_s^0$ is set as the original input gene expressions $X$. We denote $Z_s$ as the final output of the encoder, where the $i$-th row $z_i$ denotes the latent representation of spot $i$.

After that, the latent representations $Z_s$ are fed into a decoder to reverse them back into the raw gene expression space. Different from the encoder, the decoder adopts a symmetric architecture to reconstruct the gene expression. Specifically, the decoder is defined as follows,

$$H_s^t = \sigma\left(\widetilde{A} H_s^{t-1} W_d^{t-1} + b_d^{t-1}\right), \tag{2}$$

where $H_s^t$ denotes the reconstructed gene expression profiles at the $t$-th layer and $H_s^0$ is set as the output representation $Z_s$ of the encoder. $W_d$ and $b_d$ represent the trainable weight matrix and bias vector, respectively, which are shared by all nodes in the graph. To make full use of the gene expression profiles, we train the model by minimizing the self-reconstruction loss of gene expressions as follows,

$$\mathscr{L}_{recon} = \sum_{i=1}^{N_{spot}} ||x_i - h_i||_F^2, \tag{3}$$

As the output of the decoder, $H_s$ denotes the reconstructed gene expression profiles. $x_i$ and $h_i$ are the original normalized gene expression and reconstructed gene expression for spot $i$, respectively.

## Self-supervised contrastive learning for representation refinement

To make the representation $H_s$ more informative and discriminative, we further adopt a self-supervised contrastive learning (SCL) strategy to ensure that the model captures the local spatial context of spots. Specifically, with the original and corrupted graphs $G$ and $G'$ as inputs, the GNN-based encoder first generates two corresponding representation matrices $Z_s \in \mathbb{R}^{N_{spot} \times d}$ and $Z_s' \in \mathbb{R}^{N_{spot} \times d}$ for the spots. Motivated by DGI (Velickovic et al.)[53], we aggregate the neighbors' representations as the local context of a spot i, i.e., $g_i$, representing the spot's neighborhood microenvironment. Assuming that a spot in the spatial data usually contains cell type and gene expression similar to its local context, the readout function is defined by a sigmoid of the mean of the representations of immediate neighbors instead of global neighbors unlike DGI. For spot $i$ in the graph, its representation $z_i$ and the local context vector $g$ form a positive pair, while its corresponding representation $z_i'$ from the corrupted graph and the local context vector $g$ form a negative pair. The key idea behind SCL is to maximize the mutual information of positive pairs while minimizing the mutual information of negative pairs. By using contrastive learning, spatially adjacent spots will have similar representations, while nonadjacent spots will have dissimilar representations. Next, we use binary cross-entropy (BCE) to model SCL. Formally, the contrastive loss can be defined as:

$$\mathscr{L}_{SCL} = -\frac{1}{2N_{spot}}\left(\sum_{i=1}^{N_{spot}}\left(\mathbb{E}_{(X,A)}[\log\Phi(z_i,g_i)] + \mathbb{E}_{(X',A')}[\log(1-\Phi(z_i',g_i))]\right)\right), \tag{4}$$

where $\Phi(\cdot)$ is a discriminator $\mathscr{D}: \mathbb{R}^d \times \mathbb{R}^d \to \mathbb{R}$, a dual neural network that distinguishes the positive pairs from negative pairs. $\Phi(z_i,g)$ denotes the probability score that is assigned to the positive pair $(z_i,g)$. Considering that the corrupted graph $G'$ has the same topological structure as the original graph $G$, we define a symmetric contrastive loss $\mathscr{L}_{SCL\_corrupt}$ for the corrupted graph to make the model more stable and balanced,

$$\mathscr{L}_{SCL\_corrupt} = -\frac{1}{2N_{spot}}\left(\sum_{i=1}^{N_{spot}}\left(\mathbb{E}_{(X',A')}[\log\Phi(z_i',g_i')] + \mathbb{E}_{(X,A)}[\log(1-\Phi(z_i,g_i'))]\right)\right), \tag{5}$$

## Overall loss function

The representation learning module of ST data is trained by minimizing the self-reconstruction loss and contrastive loss. Briefly, the overall training loss of this module is defined as:

$$\mathscr{L} = \lambda_1 \mathscr{L}_{recon} + \lambda_2\left(\mathscr{L}_{SCL} + \mathscr{L}_{SCL\_corrupt}\right), \tag{6}$$

where $\lambda_1$ and $\lambda_2$ are the weight factors that trade-off the impacts of the reconstruction loss and the contrastive loss. Empirically, we set $\lambda_1$ and $\lambda_2$ as 10 and 1. The training of this module is independent of the next scRNA-seq and ST data integration module, and we employ the Adam optimizer[54] for the optimization. The learning rate and training epoch are set to 0.001 and 600 for both spatial clustering and multiple ST data integration tasks, while 0.001 and 1200 are used for the scRNA-seq and ST data integration task.

## Spatial domain assignment via clustering and refinement

After model training, we use the reconstructed spatial gene expression $H_s$ from the decoder (Fig. 1A) with the nonspatial assignment algorithm, mclust[55], to cluster the spots into different spatial domains. Each cluster is regarded as a spatial domain, containing spots with similar gene expression profiles and are spatially proximate. For tissue slices with manual annotation, we set the number of clusters to be the same as the ground truth. For tissue slices without manual annotations, we test different cluster counts and select the count that gives the highest Silhouette score[56]. Although the reconstructed spatial gene expression $H_s$ was obtained using both gene expression and spatial information, some spots may be wrongly assigned to spatially disparate domains. We consider such occurrences to be noise and that their presence may influence downstream biological analysis. To resolve this, we extended our model with an optional optimization step. In this step, for a given spot $i$, its surrounding spots within an $r$ radius circle are treated as its neighbors. GraphST reassigns the spot $i$ to the same domain as the most common label of its surrounding spots. Setting $r$ to 50 gave the best clustering performance in our benchmarks. This step is not recommended for ST data with fine-grained domains (e.g., mouse brain anterior and posterior) or acquired using Stereo-seq and Slide-seqV2. In this study, we only applied this refinement step to the human brain DLPFC and the human breast cancer dataset.

## Vertical and horizontal integration of multiple tissues via implicit batch effect correction

The discussion so far assumes only a single tissue slice as input. For biological analysis of tissue samples, integrated analysis of multiple tissue slices can yield greater insights. Two types of multiple sample analysis are possible, vertically split tissue slices (such as the mouse breast cancer sections 1 and 2) and horizontally split tissue slices (such as the mouse brain anterior and posterior sections). For the former, one major challenge for integrated analysis is the presence of batch effects among different slices, hindering data integration. For the latter, the challenge is to assign spots to domains such that the domains straddling the joining edge are aligned.

To overcome these challenges, we extended our GraphST model to handle the integrated analysis of multiple tissue slices. Here we consider an example with two slices, though the model can be extended to handle more slices. There are three major steps, as illustrated in Fig. 1B. First, for two given tissue slices, we employ the algorithm PASTE[57] to align their H&E images to ensure that the two slices are adjacent in space. Next, with the aligned spatial coordinates, a joint neighborhood graph for the two slices is constructed in the same way as with a single slice. The joint construction of the neighborhood graph makes it possible to consider both intra- and inter-slice adjacent spots as neighbors for a specific spot, enabling feature smoothing between adjacent spots across slices during the representation learning. Finally, with the joint neighborhood graph and gene expressions as inputs, GraphST learns the joint representation of the spots from the two slices for downstream spatial clustering (Fig. 1A).

For vertical integration, GraphST implicitly removes batch effects between slices without explicit batch factor detection. The batch effects mainly originate from the discrepancies in feature distributions between batches. In GraphST, two aspects contribute to batch effect elimination. First, GraphST learns the representation by iteratively aggregating representations of neighbors, which smooths the feature distribution of batches and helps diminish the differences between batches. Second, by using the graph self-supervised contrastive learning, the learned representation captures local context information, which further makes spatially adjacent spots have similar representations.

## PASTE (probabilistic alignment of ST experiments) for multiple tissue slice alignment

We employed PASTE[57] to align and integrate multiple tissue slices into a single consensus slice. PASTE leverages both gene expression similarity and spatial distances between spots to align and integrate spatial transcriptomics data. In our analysis, we used center slice integration to overcome variability in individual slices due to varying sequencing coverage, tissue dissection, or tissue placement on the array. We first filtered genes with min_counts = 15 in each individual slice using Scanpy. We then filtered for common genes for each individual slice and used PASTE's 'center_align' algorithm or center slice integration mode. In this mode, PASTE infers a 'center' slice consisting of a low-rank expression matrix and a collection of mappings from the spots of the center slice to the spots of each input slice and integrates the ST slices to the center slice by combining a fused Gromov–Wasserstein barycenter with nonnegative matrix factorization (NMF).

## Spatially informed contrastive learning for scRNA-seq and ST data integration

To integrate scRNA-seq and ST data, we aim to learn a trainable mapping matrix $M$ of dimensions $N_{cell} \times N_{spot}$ to project cells from the scRNA-seq data into the spatial space (Fig. 1C). Each element $M_{ij}$ in $M$ represents the probability of cell $i$ being mapped to spot $j$, with the constraint that the total probability for all the cells is 1 for each spot, i.e., $\sum_i^{N_{cell}} M_{ij} = 1$.

The mapping matrix $M$ is learned using the gene expression profiles of the scRNA-seq and ST data. Unlike Tangram[34], which learns the mapping matrix directly using the raw gene expressions with high noise levels, GraphST retains and refines the informative and noise-reduced features from the raw gene expression profiles of both the scRNA-seq and ST data via deep learning modules before learning the mapping matrix. Specifically, for ST data, we take the output $H_s$ of Module 1 (Fig. 1A) as the input for the mapping matrix learning (Fig. 1C). For the scRNA-seq data, we learn the cell representations via an auto-encoder. Specifically, with the normalized gene expression $e_i$ as input, a latent representation $q_i$ of cell $i$ is learned by an encoder:

$$q_i = f_{en}(e_i), \tag{7}$$

where $f_{en}(\cdot)$ is a multiple-layer neural network. After that, cell gene expression $y_i$ is reconstructed by a decoder:

$$y_i = f_{de}(q_i), \tag{8}$$

where $f_{de}(\cdot)$ is a multiple-layer neural network like the encoder. After model training, we can obtain the reconstructed cell gene expression matrix $H_c$, with which we can predict the spatial gene expression matrix $H_s'$ by combining it with mapping matrix $M$:

$$H_s' = M^T \cdot H_c. \tag{9}$$

To learn the mapping matrix $M$, we designed an augmentation-free contrastive learning mechanism to align the predicted spatial gene expression $H_s'$ with the reconstructed spatial gene expression $H_s$ (Fig. 1A). The overall loss $\mathscr{L}_{map}$ of the mapping matrix learning is formulated as:

$$\mathscr{L}_{map} = -\alpha \sum_{i=1}^{N_{spot}} \sum_{j \in \mathscr{N}_i} \log \frac{\exp\left(sim\left(h_i', h_j\right)/\tau\right)}{\sum_{p \neq i}^{N_{spot}} \exp\left(sim\left(h_i', h_p\right)/\tau\right)} + \beta ||H_s - H_s'||_F^2, \tag{10}$$

where $sim(i,j)$ denotes the cosine similarity of spot pair $i,j$ calculated by their representations, $\mathscr{N}_i$ is the set of neighbors of spot $i$, and $\tau$ represents temperature parameter (set as 1 by default). Here the first term computes the contrastive learning loss, aiming to maximize the similarities of positive pairs and minimize those of negative pairs. For a given spot $i$, positive pairs are defined as those that it makes with spatially adjacent spots, while negative pairs are those made with spatially nonadjacent spots. The second term is to ensure that the predicted gene expression is proportional to the reconstructed gene expression. We use $\alpha$ and $\beta$ as weight factors to control the weightage consideration of the contrastive and reconstruction losses. $\alpha$ and $\beta$ are set to 1 and 10, respectively.

Unlike most existing deconvolution methods, such as SPOTLight and cell2location, our GraphST model is independent of the scRNA-seq annotation information (e.g., cell types) during the mapping process.

## Spot-targeted annotation transfer

As the output of Module 3 (Fig. 1C), we can obtain the mapping matrix $M \in \mathbb{R}^{N_{cell} \times N_{spot}}$, which can be treated as a general transfer function. For certain annotations of scRNA-seq data, such as cell types, disease states, or disease grade, we can transfer it into the spatial space easily via $M$. Briefly, let $S_{cell} \in \mathbb{R}^{N_{cell} \times N_{annot\_cell}}$ denote a one-hot annotation matrix with rows representing cells and columns representing annotation labels, and $N_{annot\_cell}$ is the number of annotation labels. The probability distribution $P_{spot} \in \mathbb{R}^{N_{spot} \times N_{annot\_cell}}$ of cell annotations in the spots is formulated as:

$$P_{spot} = M^T \cdot S_{cell}. \tag{11}$$

To circumvent the influences of spots with low scores, we empirically retain the scores of the top 10% cells for each spot in $P_{spot}$ and set the remaining values to 0.

## Domain-targeted annotation transfer

Although we can determine the spatial distribution of annotations (taking cell type as an example) in the H&E images, some cell types may span different spatial domains such that it becomes difficult to distinguish which cell types are found in which spatial domains. Therefore, we further extend our model to transfer annotations of scRNA-seq data into the spatial domain. With function (11), we can derive the annotation-to-spot projection matrix $P_{spot} \in \mathbb{R}^{N_{spot} \times N_{annot\_cell}}$. We assume that $S_{spot} \in \mathbb{R}^{N_{spot} \times N_{annot\_spot}}$ represents a one-hot annotation

matrix with $N_{annot\_spot}$ denoting the number of spatial domains. Similar to the spot-targeted annotation transfer, the probability distribution $P_{domain} \in \mathbb{R}^{N_{annot\_spot} \times N_{annot\_cell}}$ of cell annotations in spatial domains is formulated as:

$$P_{domain} = S_{spot}^T \cdot P_{spot}.$$ (12)

## Mouse breast cancer experiments

All animal work was approved by the NUS Institutional Animal Care and Use Committee (IACUC) and was in accordance with the National Advisory Committee for Laboratory Animal Research (NACLAR) Guidelines (Guidelines on the Care and Use of Animals for Scientific Purposes) (Protocol approval R18-0635). Here, $1 \times 10^5$ metastatic murine breast cancer cells (4T1) tagged with luciferase were implanted into the mammary fat pads of 16- to 20-week-old Balb/cNTac mice. Tumor growth was monitored over 2 weeks using a digital caliper. Tumors were excised and extracted at 2 weeks for spatial transcriptomics.

## Histology and RNA quality assessment

4T1 implanted mammary pad tumors during posttreatment were excised, immediately embedded in optimal cutting temperature (OCT) compound, and frozen on dry ice. Cryoblocks were kept at −80 °C and sent for histology analysis to obtain 20-μm sections. RNA quality was assessed for all tissue blocks using the RNeasy Mini kit (Qiagen). Ten sections were obtained from each cryoblock to verify the RNA quality for each respective block. Then, 600 μl of buffer RLT was added to the 10 sections and subsequently disrupted using a QIAshredder by centrifuging for 2 mins at maximum speed. RNA was extracted from the lysate with the RNeasy Mini kit using instructions from the "Purification of Total RNA from Animal Tissues" section, and quality was assessed via the RNA integrity number (RIN) value determined using the Agilent 2100 Bioanalyzer (Agilent). Cryoblocks-derived sections with RIN value ≥ 8 were used for the subsequent spatial transcriptomics experiments.

## Spatial transcriptomics

Sections obtained at 20 μm were placed within the grids on pre-chilled Visium Tissue Optimization and Gene Expression slides (10x Genomics) and stored at −80 °C. For H&E staining, slides with sections were thawed at 37 °C for 1 min before fixation in methanol at −20 °C for 30 mins. H&E staining was performed according to the manufacturer's protocol (10x Genomics; CG000160), with hematoxylin staining reduced from 7 mins to 5.5 mins. The optimal tissue permeabilization time of individual cryoblocks of each group was obtained from time-course assays performed on tissue optimization slides. This optimal timing was then used on the gene expression slides with the respective blocks to capture mRNA post-permeabilization. Images were all obtained and stitched together using EVOS FL Auto 2 (Thermofisher Scientific) with a 20x objective (Fluorite with correction collar; 20x objective, 0.7 N.A), and the raw images were saved in the TIF format. Fluorescence images were acquired using an RFP filter cube (531/40 nm Ex; 593/40 nm Em) with a 20x objective (Plan fluorite). The created RNA-seq libraries were then sequenced with Novaseq.

## Data description

For spatial clustering, we employed five spatial gene expression datasets (see Supplementary Table S1 for details). The first dataset was the LIBD human dorsolateral prefrontal cortex (DLPFC) with 12 tissue slices acquired with 10x Visium[36] (http://research.libd.org/spatialLIBD/). The number of spots in each slice ranged from 3460 to 4789, with 33,538 genes captured. Each slice was manually annotated to contain five to seven regions, namely the DLPFC layers and white matter. The second dataset of mouse brain tissue was downloaded from the publicly available 10x Genomics Data Repository (https://www.10xgenomics.com/resources/datasets). This dataset had two sections, of which we selected the anterior section. The selected section contained 2695 spots with 32,285 genes captured and was manually annotated with 52 regions using the Allen Brain Atlas reference (https://mouse.brain-map.org/static/atlas). The third dataset of the mouse olfactory bulb was acquired using Stereo-seq[22], which was further processed and annotated[12,58]. The data contained 19,109 spots and 14,376 genes. The fourth dataset of the mouse hippocampus was acquired with Slide-seqV2 and was downloaded from (https://portals.broadinstitute.org/single_cell/study/slide-seq-study). The section used was Puck_200115_08, with 52,869 spots. The last dataset consisted of Stereo-seq data acquired from two mouse embryos (E9.5 and E14.5)[22], which we downloaded from https://db.cngb.org/stomics/mosta/. The E9.5 embryo data consisted of 5913 bins and 25,568 genes, while the E14.5 embryo data consisted of 92,928 bins and 18,566 genes.

In the second task of multiple sample integration, we performed vertical integration on two datasets and one for horizontal integration (see Supplementary Table S1 for details). For vertical batch integration, both datasets were acquired from mouse breast cancer tissue that is described above. We utilized two sets of samples derived from this tissue, of which each set is composed of two vertically adjacent sections. The number of spots ranged from 1868 to 3042 for each section, with 32,285 genes captured. For horizontal integration, we employed both sections (anterior & posterior) of the mouse brain tissue dataset that was also used for clustering.

In the third task of ST cell composition deconvolution with scRNA-seq reference data, we employed both simulated and experimentally acquired data. The first example tested simulated data obtained from the benchmarking study by Li et al.[41], downloaded from https://github.com/QuKunLab/SpatialBenchmarking/tree/main/FigureData/Figure4. We used datasets 4 and 10 from the study. Dataset 4 was generated from mouse cortex experimental data acquired with seqFISH+ and Smart-seq, with 72 spots created. Dataset 10 was generated using mouse visual cortex tissue data captured with STARmap and Smart-seq technologies, with 189 spots created. For experimentally acquired datasets, we used four examples (see Supplementary Tables S1 and S2 for details). The first example consisted of human lymph node tissue data acquired with spatial transcriptomics and scRNA-seq, both obtained from an existing study by Kleshchevnikov et al.[29]. The ST data contained 4035 spots with 36,588 genes while the scRNA-seq consisted of 73,260 cells with 10,217 genes captured. The second example employed the anterior section of the mouse brain sample data. The scRNA-seq reference used was acquired from mouse whole cortex and hippocampus tissues, encompassing more than 1.1 million cells with 22,764 genes captured (https://portal.brain-map.org/atlases-and-data/rnaseq/mouse-whole-cortex-and-hippocampus-10x). The third example used the human brain sample DLPFC dataset, of which we used slice #151673. The corresponding snRNA-seq data of archived postmortem dorsolateral prefrontal cortex (BA9) tissue was acquired with the 10x Genomics Chromium platform (https://www.ncbi.nlm.nih.gov/geo/query/acc.cgi?acc=GSE144136). The snRNA-seq dataset was composed of 78,886 cells with 30,062 genes[46]. The final deconvolution example utilized the human breast cancer sample obtained from the publicly available 10x Genomics Data Repository (https://www.10xgenomics.com/resources/datasets/human-breast-cancer-block-a-section-1-1-standard-1-1-0). The dataset contained 3798 spots with 36,601 genes and was manually annotated with 20 regions. The corresponding scRNA-seq data were downloaded from the database DISCO[47], consisting of cells from two different samples, i.e., adjacent normal and solid tumor (https://www.immunesinglecell.org/). The data consisted of 476,571 cells with 5000 genes, which we downsampled to 46,080 cells while maintaining the relative cell-type composition.

## Comparison with baseline methods and evaluation

To showcase the effectiveness of GraphST in spatial clustering, we compared GraphST with seven state-of-the-art methods, the non-spatial method Seurat[7] and spatial methods Giotto[8], SpaGCN[9], SpaceFlow[14], conST[15], BayesSpace[11], and STAGATE[12]. To evaluate the performance of multiple tissue slice integration, we compared GraphST with Harmony[16], scVI[17], and STAGATE for vertical integration and STAGATE and SpaGCN for horizontal integration. For ST cell-type deconvolution with scRNA-seq data, GraphST was compared with cell2location, which had been benchmarked to be the best existing method[41], as well as Seurat, RCTD, SPOTlight, and NNLS.

*Seurat.* Seurat is a popular single-cell transcriptomics analysis library that has been extended to handle spatial transcriptomics data. We used the default pipeline and default parameters in Seurat as described in the spatial clustering vignette. As we were unable to specify the number of clusters in Seurat, we ran the FindClusters function at different resolutions and chose the resolution that gave us the desired number of clusters.

*Giotto.* Giotto is a toolbox designed for spatial data analysis. We followed the tutorial in Giotto's Github repository with default settings to preprocess the data. For clustering, we used the hidden random Markov random field model-based module designed for spatial pattern discovery. Specifically, we used the Giotto functions, init_hmrf_v2 and do_hmrf_v2 with default parameter values. The HMRF workflow is given at: https://bitbucket.org/qzhudfci/smfishhmrf-r/src/master/TRANSITION.md.

*SpaGCN.* SpaGCN is a graph convolutional network approach that integrates gene expression, spatial location information, and histological images for ST data analysis. SpaGCN is one of the only two other methods that can perform horizontal ST data integration. Following the tutorial, we applied SpaGCN to spatial clustering and horizontal ST data integration with the default parameter settings. In particular, the parameter 'histology' was set to 'False'. The learning rate and max training epoch were set to 0.05 and 200, respectively.

*conST.* conST is a contrastive learning-based method that combines gene expression, spatial information, and morphology for ST data analysis. We applied conST on the DLPFC samples for spatial clustering with default parameter settings. Specifically, the parameter 'use_img' was set to 'False'. The number of neighbors was set to 10 when constructing the neighborhood graph. The training epoch and learning rate were set to 200 and 0.01, respectively.

*SpaceFlow.* SpaceFlow is a deep graph network that is developed for ST data analysis by combining gene expression profiles with their spatial locations. We ran SpaceFlow for comparison on spatial clustering. We used the default parameter settings to preprocess data and run the model. In particular, the gene with expression in fewer than three cells and cells with expression of fewer than 100 genes were removed. After that, the normalized expression was multiplied by a scale factor of 10,000 and log-transformed with a pseudo-count one. As feature inputs, 3000 highly variable genes were selected. The learning rate, training epoch, and regularized weight factor were set at 0.001, 1000, and 0.1, respectively.

*BayesSpace.* BayesSpace uses a Bayesian model with a Markov random field to model spatial transcriptomics data for clustering, utilizing both spatial and gene expression information. We followed the analysis tutorial for BayesSpace in its GitHub repository and used the following parameters, nrep = 50,000 and gamma = 3, platform = "Visium" and mode = "normal".

*STAGATE.* STAGATE is another deep learning model-based method that combines an auto-encoder with a graph attention mechanism to learn latent representation by modeling both gene expression profiles and spatial location information. We ran STAGATE for spatial clustering and vertical and horizontal ST data integration. All experiments were implemented using the recommended parameters in the package vignette. Specifically, with raw gene

expressions, the top 3000 highly variable genes were first selected and then log-transformed and normalized according to library size. The parameter 'alpha' was set to 0. The learning rate and training epoch were left at the default 0.0001 and 500, respectively.

*Harmony.* Harmony is a nonspatial batch correction method. We used Harmonypy (https://github.com/slowkow/harmonypy) for batch integration of the two mouse breast cancer datasets. The dataset was first preprocessed using the standard Scanpy workflow, including log-normalization, scaling, and PCA dimension reduction. We used the PCA embeddings and sample batch information as input to Harmonypy and obtained the batch-corrected embeddings.

*scVI.* scVI is another nonspatial batch correction method that combines stochastic optimization with deep neural networks. We employed the scVI package (version 0.19.0) for batch integration. The input dataset was preprocessed using the standard Scanpy workflow. After log-normalization, 1200 highly variable genes were selected as input for data integration. The model was set up and trained using the following default parameters: 'n_hidden: 128, n_latent: 10, n_layers: 1, dropout_rate: 0.1, dispersion: gene, gene_likelihood: zinb, latent_distribution: normal'. The output of corrected latent embeddings was then used for downstream analysis.

*cell2location.* cell2location employs a Bayesian model to estimate the spatial distribution of cell types in the ST data of a given tissue using single-cell or nuclei RNA-seq data as reference. We first performed the initial preprocessing of removing mitochondrial genes from the ST data. For the scRNA-seq reference dataset, we performed very permissive gene selection using the following parameters: 'cell_count_cutoff = 5'; 'cell_percentage_cutoff = 0.03' and 'nonz_mean_cutoff = 1.12'. We used these selection criteria instead of the standard highly variable gene selection to retain rare marker genes while removing most of the uninformative genes. To estimate reference cell-type signatures from the single-cell RNA-seq profiles, we used cell2location with the default negative binomial regression. We used the following parameters: 'max_epochs = 250'; 'batch_size = 2500'; 'train_size=1'; 'lr=0.002'; 'use_gpu=True' to train the model and export the estimated cell abundance for each cell type. In the next step, cell2location performed spatial mapping by taking the spatial dataset and estimated cell abundance for each cell type from scRNA-seq reference dataset as input to output the estimated cell abundance at all spatial locations. Here, we set the hyperparameters: 'N_cells_per_location=30' (number of cells per location) and 'detection_alpha=20' (for within-experiment variation in RNA detection) and other parameters such as: 'max_epochs=30000'; 'batch_size=None'; 'train_size=1' and 'use_gpu=True'. The cell abundance in each spatial location was visualized in scatter plots with Scanpy's 'scanpy.pl.spatial' command.

*RCTD.* RCTD is a model-based approach that leverages cell type profiles learned from single-cell RNA-seq to decompose cell type mixtures. We followed the guidelines on the RCTD GitHub repository: https://raw.githack.com/dmcable/spacexr/master/vignettes/spatial-transcriptomics.html. The model was set up and trained using the following parameters: doublet_mode = 'full'.

*SPOTlight.* SPOTlight is a nonnegative matrix factorization regression model that uses a modified nonnegative least squares (NNLS) method to deconvolute ST capture locations (spots). We followed the guidelines on the SPOTlight GitHub repository: https://marcelosua.github.io/SPOTlight/. The model was set up and trained using the following parameters, transf = 'uv', method = 'nsNMF'.

*NNLS.* (AutoGeneS) NNLS is a computational method to estimate the absolute cell abundance of cell types given the reference signatures of cell types. NNLS implementation is a part of the AutoGeneS package.

## Statistics and reproducibility

No statistical method was used to predetermine the sample size. No data were excluded from the analyses. The experiments were not

randomized. The investigators were not blinded to allocation during experiments and outcome assessment.

## Reporting summary

Further information on research design is available in the Nature Portfolio Reporting Summary linked to this article.

## Data availability

Datasets analyzed in this paper are available in raw form from their original authors (see Supplementary Tables S1 and S2). The data used in this study have been uploaded to Zenodo and is freely available at: https://zenodo.org/record/6925603#.YuM5WXZBwuU.

## Code availability

An open-source Python implementation of the GraphST toolkit is accessible at https://github.com/JinmiaoChenLab/GraphST.

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

## Acknowledgements

The work was supported by AI, Analytics and Informatics (AI3) Horizontal Technology Programme Office (HTPO) seed grant (Spatial transcriptomics ST in conjunction with graph neural networks for cell–cell interaction #C211118015) from A * STAR, Singapore; Open Fund Individual Research Grant (Mapping hematopoietic lineages of healthy and high-risk acute myeloid leukemia patients with FLT3-ITD mutations using single-cell omics #OFIRG18nov-0103) from Ministry of Health, Singapore; Singapore National Research Foundation grant #NRF-CRP19-2017-04; Industry Alignment Fund (Pre-Positioning) grant (SinGapore Immu-NogrAm for ImmunoOncoLogy #IAF-PP H19/01/a0/024) from the National Research Foundation, Singapore; the National Research Foundation, Singapore, and Singapore Ministry of Health's National Medical Research Council under its Open Fund-Large Collaborative Grant ("OF-LCG") (#MOH-OFLCG18May-0003); Singapore National Medical Research Council (#NMRC/OFLCG/003/2018); Singapore A*STAR Central Research Fund.

## Author contributions

J.C. conceptualized and supervised the project. Y.L. designed the model with feedback from H.F. and M.W. Y.L. developed the GraphST software. Y.L., K.S.A., and J.C. wrote the manuscript. Y.L., K.S.A., and J.C. led the data analysis with input from M.L., K.L.K.C., R.S., C.Z., and H.X. Y.L. contributed to figure design and generation. Z.O. and L. Z. annotated and interpreted the brain datasets. K.L.K.C. performed the 10x Visium experiments to generate the mouse breast cancer spatial transcriptomic data. K.S. and L.H.K.L. performed the mouse experiments, provided the breast cancer tissues, and annotated the ST data. A.C. and L.L. provided the Stereo-seq data of the mouse olfactory bulb.

## Competing interests

The authors declare no competing interests.
