## [Peer Review File · Nature Communications]

Reviewers' Comments:

Reviewer #1:

Remarks to the Author:

In this manuscript, Long et al. present a deep graph network-based method DeepST for 1) inference of spatial domains, 2) data integration between ST samples or between ST and scRNA-seq data, 3) deconvolution. The method utilized graph neural network together with self-contrastive learning techniques. The authors showed better performance than several benchmarked methods, and demonstrated its abilities on identifying meaningful spatial domains and integrating different samples or data modalities.

Overall, the topic is important and timely. While this method has strong potential in application, the manuscript lacks clarity on the details of the method, and it is unclear what unique advantages the method has compared with other existing methods in terms of its performance. Here are several specific major points in addition to multiple minor point for improving the work. Below are detailed comments:

Major concerns:

1. Lines 123-124, "To our knowledge, few existing methods use self-supervised contrastive learning on spatial transcriptomics". While the work may be new in using self-supervised contrastive learning on spatial transcriptomics, there are some papers that use such kind of methods to address similar questions. For example,
 - 1) Ren, H., Walker, B.L., Cang, Z. et al. Identifying multicellular spatiotemporal organization of cells with SpaceFlow. Nat Commun 13, 4076 (2022). <https://doi.org/10.1038/s41467-022-31739-w>
 - 2) conST: an interpretable multi-modal contrastive learning framework for spatial transcriptomics. Yongshuo Zong, Tingyang Yu, Xuesong Wang, Yixuan Wang, Zhihang Hu, Yu Li bioRxiv 2022.01.14.476408; doi: <https://doi.org/10.1101/2022.01.14.476408>.
What is the novelty of the proposed method compared to those two methods?
2. I tried to test and reproduce results shown in this manuscript using the provided links in the manuscript (<https://deepst-tutorials.readthedocs.io/>), and observed the following problem:
 - a. DeepST/utlils.py:121, in refine_label(adata, radius, key)

```
120 for j in range(1, n_neigh+1):  
 121 neigh_type.append(old_type[index[j]])
```

IndexError: index 3602 is out of bounds for axis 0 with size 3583

Basically, in the clustering step (In refine_label function), the shape of old_type and index are inconsistent, which might be caused by the inconsistent size between the dimensions from adata and distance matrix. As a result, I couldn't reproduce the results shown in the paper. See the attached `DeepST_test.ipynb` for details.

3. Regarding to the method, especially the contrastive learning component, both the formula and the ideas are very similar to Deep Graph Infomax (DGI) (Veličković et al. 2018). What are the novel elements and major differences between current method and DGI? This needs to be addressed.
4. What are the meaning and motivation of formula (5), line 734? It's important to show the performance difference with and without adding this term by experiment, because DGI only contains (4) instead of (5). Does this term actually improve the performance?
5. The reason of using reconstructed expression data to cluster instead of using the latent embedding need to be justified. Moreover, why choosing mclust over graph-based methods such as Leiden, Louvain? It's important to justify such choices in terms of data analysis and results.
6. In line 684, do the authors augment data through creating corrupted graph by randomly adding or dropping edges? What is the effect of such procedure on the overall performance of the method.
7. Regarding the data integration performance shown in Figure 4, why did the authors not compare many other methods designed for nonspatial scRNA-seq data, such as scVI (Lopez et al. 2018) and Harmony (Korsunsky et al. 2019), because those classical methods have been well demonstrated for good performance for single-cell data.
8. It's important to show the STAGATE results that similar to Fig 4E to better demonstrate the data

integration performance.

Minor points

9. To better support the manual annotation result in Fig 6A, the spatial expression distribution of several marker genes for each panels in Fig 6A need to be added.
10. In line 836, the author mentioned the first loss term indicates contrastive loss, why is there only one instead of two terms? What is the meaning of the first term?
11. In line 712, which norm is used? L1 or L2 or others?
12. In Fig 3C, the titles of panels Mesenchyme and Dermomyotome seem misplaced.
13. All color bars need to be explained for their meanings.
14. Many typos and grammar errors in the manuscript, e.g., in line 28, "has" should be "have"; in line 59, "K-means" should be "k-means"
15. Lines 124-126, "Using self-supervised contrastive learning improves performance in learning relevant latent features and has the additional benefit of removing batch effects". This sentence occurs without any supporting evidence. It needs to be fixed.
16. In lines 158-162, the authors introduced "self-reconstruction loss" and "contrastive loss" and their effects. It's important to show what the two losses are in the context of biology.
17. It's unclear how the neighbor graph is constructed. In the caption of Fig. 1 (lines 1034-1035), the authors wrote "...neighbor graph constructed using spot coordinates (x,y) of that fall within a distance threshold". However, in the method section in lines 665-675, the authors wrote "Finally, we select the top k-nearest spots as its neighbors". It's unclear whether the authors used a distance threshold or a threshold for k.
18. Regarding the method (lines 655-659), the descriptions seem to be for the spatial transcriptomics data. However, this is not clear from the description, as two kinds of datasets (spatial transcriptomics data and scRNA-seq data) are mentioned in this paper.
19. In lines 684-688, "...while keeping the original graph structure unchanged": was the corrupted neighbor graph G' the same as the original G ?
20. In lines 708-709, " W_d and b_d represent the trainable weight matrix and bias vector, respectively, which are shared by all nodes in the graph". Please justify why W_d and b_d need to be shared by all nodes in the graph. Besides, is this the same case for W_e and b_e ?

Reference

- Korsunsky, Ilya, Nghia Millard, Jean Fan, Kamil Slowikowski, Fan Zhang, Kevin Wei, Yuriy Baglaenko, Michael Brenner, Po-Ru Loh, and Soumya Raychaudhuri. 2019. "Fast, Sensitive and Accurate Integration of Single-Cell Data with Harmony." *Nature Methods* 16 (12): 1289–96.
- Lopez, Romain, Jeffrey Regier, Michael B. Cole, Michael I. Jordan, and Nir Yosef. 2018. "Deep Generative Modeling for Single-Cell Transcriptomics." *Nature Methods* 15 (12): 1053–58.
- Veličković, Petar, William Fedus, William L. Hamilton, Pietro Liò, Yoshua Bengio, and R. Devon Hjelm. 2018. "Deep Graph Infomax." *ArXiv [Stat.ML]*. arXiv. <http://arxiv.org/abs/1809.10341>.

Reviewer #2:

Remarks to the Author:

In the manuscript "DeepST: A novel graph" by Long and coworkers the authors develop a new method for better describing spatial transcriptomics data and being able to integrate multiple studies. The method is based on graph neural networks and contrastive learning, which makes it possible to combine scRNA-seq of better resolution and spatial transcriptomics. The method is logically sound and makes a lot of sense. Moreover, authors show that it empirically identifies more relevant clusters and allows data integration for higher power. Although, I am not an expert in spatial transcriptomics these problems seem of great importance and authors spend good effort to show that it works a planned.

Having said that my expertise is in neural networks and translational bioinformatics I believe that the paper would be a good contribution to the spatial transcriptomics field. From my side, I have no concerns of the paper and like to see it published.

**Response to comments for paper NCOMMS-22-35863B “Spatially informed clustering,**
**integration, and deconvolution of spatial transcriptomics with GraphST”**

**Response to Reviewer #1**

**Summary:** *In this manuscript, Long et al. present a deep graph network-based method GraphST for 1)*
*inference of spatial domains, 2) data integration between ST samples or between ST and scRNA-seq*
*data, 3) deconvolution. The method utilized graph neural network together with self-contrastive learning*
*techniques. The authors showed better performance than several benchmarked methods, and*
*demonstrated its abilities on identifying meaningful spatial domains and integrating different samples or*
*data modalities.*

*Overall, the topic is important and timely. While this method has strong potential in application, the*
*manuscript lacks clarity on the details of the method, and it is unclear what unique advantages the*
*method has compared with other existing methods in terms of its performance. Here are several specific*
*major points in addition to multiple minor point for improving the work. Below are detailed comments:*

**Response:** We thank the reviewer for the positive comments. We summarize Graph ST’s technological
novelty and advantages over existing methods in the following paragraphs. We have carefully
addressed all the comments and suggestions when preparing this revision of the manuscript. Please
let us know if you have additional comments.

With rapid technological advances in spatial transcriptomics, it is now widely applied towards studying
tissue complexity and cell-cell communications. However, the current bottleneck still lies in data
analysis. Although multiple methods have been developed for spatial transcriptomics, there is still a
great need for developing novel tools that offer greater accuracy, robustness, and generalizability
towards a wide range of application on different tissue types and technology platforms. Furthermore,
the analysis pipeline for spatial transcriptomic data comprises three key tasks, namely spatial clustering,
multi-sample integration, and cell type deconvolution. However, there is no comprehensive tool that can
perform all these three tasks. To overcome this limitation, we developed GraphST, the first of its kind
that integrates these tasks into a streamlined process. Most importantly, GraphST outperforms existing
methods in each task. We achieved this by adopting and tailoring graph self-supervised contrastive
learning for spatial transcriptomics analysis.

In the spatial clustering task, we achieved higher accuracy and robustness with an average of 10%
improvement over the best of existing methods in a variety of datasets. Our GraphST clusters revealed
finer tissue structures and niches in complex tissues such as the brain, olfactory bulb, and embryo.
Although existing methods conST and SpaceFlow also adopted graph contrastive learning for spatial
clustering, there are major technical differences and performance advantages when comparing
GraphST to conST and SpaceFlow. Briefly, GraphST is different from DGI, conST and SpaceFlow in
three aspects: A) definition of positive/negative pairs, B) objective function and contrastive loss, and C)
training procedure. These differences enable GraphST to outperform the other methods in the spatial
clustering task. Furthermore, we have conducted several ablation studies to confirm that each of these
differences improves the effective integration of gene expression and spatial context to obtain
informative and discriminative latent representations. Please kindly refer to Response 1.1 for details of
comparison between GraphST and conST, SpaceFlow.

In the multi-sample integration task, GraphST can better correct batch effects when integrating serial
tissue slices than existing methods that have been developed for spatial (e.g., STAGATE) or non-spatial
batch integration (e.g., scVI and Harmony). Moreover, for the horizontal integration of mouse anterior
and posterior brain slices, GraphST outperformed SpaGCN and STAGATE in that GraphST could
assign the common cortical layers that aligned well across the shared edge and also reveal the dorsal
and ventral horns of the hippocampus regions.

In the final task, GraphST produced more accurate cell type deconvolution with simulation data than
existing methods, including cell2location that was recognized as the top performing method in a recent
benchmark. Moreover, when applied to 10x Visium acquired human lymph node data, GraphST’s

deconvolution was able to better capture the germinal centers and mapped the B cell subpopulations
with higher spatial coherence. Lastly, application on human breast cancer 10x Visium data revealed
immune cell distributions across healthy, tumor edge, invasive ductal carcinoma (IDC), and ductal
carcinoma in situ (DCIS) regions. In particular, the T cells enriched in the IDC regions showed
upregulation of known exhaustion markers including *LAG3*, *TIGIT*, *PD1*, *TIM3*, and *CTLA4*, suggesting
a tumor induced immune suppressive environment.

**Major concerns:**

**Comment 1.1.** Lines 123-124, “To our knowledge, few existing methods use self-supervised contrastive
learning on spatial transcriptomics”. While the work may be new in using self-supervised contrastive
learning on spatial transcriptomics, there are some papers that use such kind of methods to address
similar questions. For example,

1) Ren, H., Walker, B.L., Cang, Z. et al. Identifying multicellular spatiotemporal organization of cells with
SpaceFlow. *Nat Commun* 13, 4076 (2022). <https://doi.org/10.1038/s41467-022-31739-w>

2) conST: an interpretable multi-modal contrastive learning framework for spatial
transcriptomics. Yongshuo Zong, Tingyang Yu, Xuesong Wang, Yixuan Wang, Zhihang Hu, Yu Li
*bioRxiv* 2022.01.14.476408; doi: <https://doi.org/10.1101/2022.01.14.476408>.

*What is the novelty of the proposed method compared to those two methods?*

**Response 1.1:** Thank you very much for raising this critical issue. Indeed, GraphST bears much
similarity to conST and SpaceFlow with the use of graph contrastive learning for spatial clustering.
However, there are several major differences between GraphST and the other two methods.

First, and most importantly, both conST and SpaceFlow were mainly developed for spatial clustering
only. In addition to spatial clustering, GraphST can be also applied to two other important ST data
analysis tasks, multi-sample integration and cell type deconvolution of ST. GraphST comprises three
modules with different network architectures tailored for each of the three tasks respectively.

Secondly, even for the spatial clustering task, there are also major differences when comparing
GraphST to conST and SpaceFlow, despite all three methods adopting graph contrastive learning
similar to DGI. Here we elaborate on their differences in three aspects: A) definition of positive/negative
pairs, B) objective function and contrastive loss, and C) training procedure.

82 A) GraphST’s contrastive learning is different from DGI, conST, and SpaceFlow in terms of their
definition of positive/negative pairs. DGI, conST, and SpaceFlow construct positive/negative pairs
by pairing spot embedding h_i/h'_i from the original/corrupted graph with a global summary vector
s_{global} (as shown in Figure R1 (a)). Therefore, the spot embedding learned by DGI, conST, and
SpaceFlow captures more of the global structure information but less spot-specific local
neighbourhood information. Such contrastive learning may result in feature overfitting and reduced
spot-to-spot variability. To deal with this issue, GraphST improves over DGI’s contrastive learning
by re-defining the positive/negative pairs. Specifically, motivated by the assumption that different
spots in a tissue sample have different local spatial contexts, we define positive/negative pairs by
pairing spot embedding h_i/h'_i with its local summary vector s_{local} (as shown in Figure R1(b)) instead
of the global summary vector. With the local summary vector, the model can better preserve local
context information and spot-to-spot variability. We demonstrate the effectiveness of local context
with an ablation study describe in Figure R4.

Figure R1. Illustrations of local and global summary vectors.

B) GraphST is also different from SpaceFlow, conST, and DGI in terms of the objective function and
contrastive loss formulations. The objective function of GraphST includes contrastive loss and
reconstruction loss, while DGI's objective function includes only contrastive loss. The objective
function of SpaceFlow includes both contrastive loss and a spatial consistency penalty term.
Addition of the penalty term helps SpaceFlow bring spatially adjacent spots closer in the latent
embedding. However, the lack of reconstruction loss in DGI and SpaceFlow may lead to insufficient
preservation of the original gene expression information. In contrast, GraphST adds reconstruction
loss to its objection function to ensure that the latent embedding preserves the original gene
expression information effectively. Furthermore, the contrastive loss functions are also different
between GraphST, DGI, and SpaceFlow. GraphST adopts symmetric contrastive loss (formulas (4)
and (5)) for model training while conST and SpaceFlow use single contrastive loss like DGI.
Symmetric contrastive loss can help stabilize the model and learn a better representation as
illustrated in Figure R5.

C) Lastly, although conST's objective function also contains contrastive loss and reconstruction loss
like GraphST, GraphST's training procedure is different from that of conST. conST splits the training
into pre-training and major training stages, where the model is trained with reconstruction loss in
the pre-training stage and contrastive loss in the major training stage. This two-stage training
procedure lacks mutual constraints on the contrastive and reconstructive loss, thus may fail to
identify the optimal combination of the two loss functions. In contrast, GraphST trains the model in
a single step by jointly optimizing for the reconstruction and contrastive losses. During this training,
GraphST can adaptively adjust the contributions of the different loss functions to achieve better
representation learning.

Based your comments, we compared the performance of conST, SpaceFlow, and GraphST in the
spatial clustering task with the DLPFC dataset. Figure R2 (A) shows the median ARI scores of the
different methods. We can see that GraphST achieves a much higher median ARI score of 0.60 over
conST (0.41) and SpaceFlow (0.41). Figure R2 (C) shows the results of SpaceFlow, conST, and
GraphST on slice #151673. Visually, SpaceFlow has the poorest performance among the three
methods. The domains identified by SpaceFlow are irregular though it can accurately recover the WM
domain and layer 1. conST performs slightly better than SpaceFlow with each identified domain being
continuous. However, most of the domains do not match the manual annotation well. In contrast,
GraphST's clusters are more continuous than conST and SpaceFlow, and are more consistent with the
manual annotation. Quantitatively, GraphST achieves the highest ARI score of 0.64 among the three
methods. Overall, GraphST outperforms conST and SpaceFlow in spatial clustering. The results of all
12 DPLFC slices are shown in Figure R3, which again illustrates GraphST's advantages over conST
and SpaceFlow. We have added these results to Figure 2 in the revised manuscript and Supplementary
Figure S1.

To evaluate the effectiveness of local context over global context, we conducted an ablation study by
 comparing GraphST with a variant that uses a global summary vector instead of local summary vectors.
 We ran GraphST and the variant on the 12 DLPFC slices and evaluated their performance using their
 median ARI scores. Figure R4 shows that GraphST outperforms the variant (median ARI score of 0.51)
 with a significantly higher median ARI score of 0.60. This ablation study demonstrated that local context
 does help GraphST perform better than with the global context. We have added these results to
 Supplementary Figure S14A in the revised manuscript.

To demonstrate the advantage of symmetric contrastive loss over single contrastive loss, we conducted
 another ablation study to compare GraphST with a variant that does not use the contrastive corrupted
 loss (formula (5) in the manuscript). We tested GraphST and this variant on the 12 DLPFC samples
 and evaluated their performance with the ARI metric. Figure R5 shows that GraphST achieved much
 better performance than the variant, showing that the contrastive corrupted loss contributes to better
 embedding learning. We have added these results to Supplementary Figure S14B of the revised
 manuscript.

Figure R2. Comparison between SpaceFlow, conST and GraphST on DLPFC.

Manual annotation

Figure R3. Comparison between SpaceFlow, conST, and GraphST on DLPFC.

Figure R4. Comparison analysis between GraphST and the variant that uses a global summary vector
 on DLPFC.

Figure R5. Comparison analysis between GraphST and its variant GraphST without contrastive
 corrupted loss on DLPFC.

**Comment 1.2.** I tried to test and reproduce results shown in this manuscript using the provided links in
 the manuscript (<https://GraphST-tutorials.readthedocs.io/>), and observed the following problem:
 a. `GraphST/utills.py:121, in refine_label(adata, radius, key)`

`120 for j in range(1, n_neigh+1):`

` 121 neigh_type.append(old_type[index[j]])`

`IndexError: index 3602 is out of bounds for axis 0 with size 3583`

Basically, in the clustering step (In `refine_label` function), the shape of `old_type` and `index` are
 inconsistent, which might be caused by the inconsistent size between the dimensions from `adata` and
 distance matrix. As a result, I couldn't reproduce the results shown in the paper. See the attached
 `GraphST_test.ipynb` for details.`

**Response 1.2:** We are sorry for this error. We have revised and updated our codes that are openly
 accessible (<https://github.com/JinmiaoChenLab/GraphST>). Furthermore, we provide a detailed tutorial
 to guide users on using our tool. The tutorial is available at [https://deepst-
 tutorials.readthedocs.io/en/latest/](https://deepst-tutorials.readthedocs.io/en/latest/). We welcome you to test our codes again according to the tutorial.

**Comment 1.3.** Regarding to the method, especially the contrastive learning component, both the
 formula and the ideas are very similar to Deep Graph Infomax (DGI) (Veličković et al. 2018). What are
 the novel elements and major differences between current method and DGI? This needs to be
 addressed.

**Response 1.3:** Thank you for the comments. As discussed above, the main differences between
 GraphST and DGI include the definition of positive/negative pairs and the contrastive loss functions,
 which we elaborate in the following paragraphs. In addition to the contrastive loss, GraphST employs
 reconstruction loss to preserve the original gene expressions in the latent embedding. DGI in contrast
 only uses contrastive loss.

DGI constructs positive/negative pairs by pairing each spot embedding h_i/h'_i from the original/corrupted
 graph with a global summary vector S_{global} (as shown in Figure R1 (a)). Therefore, the spot embedding
 learned by DGI captures more of the global structure information but less spot-specific local
 neighbourhood information. Such contrastive learning may result in feature overfitting and reduced
 spot-to-spot variability. To deal with this issue, GraphST improves over DGI's contrastive learning by
 re-defining the positive/negative pairs. Specifically, motivated by the assumption that different spots in
 a tissue sample have different local spatial contexts, we define positive/negative pairs by pairing each
 spot embedding h_i/h'_i with its local summary vector S_{local} (as shown in Figure R1(b)) instead of the
 global summary vector. With local summary vectors, the model can better preserve local context
 information and spot-to-spot variability. To evaluate the effectiveness of using local context over the
 global context, we conducted an ablation study by comparing GraphST with a variant that uses the

global summary vector instead of local summary vectors. We ran GraphST and the variant on the 12
 DLPFC slices and evaluated their performance with their median ARI scores. Figure R4 shows that
 GraphST outperformed the variant (median ARI score of 0.51) with a significantly higher median ARI
 score of 0.60. This demonstrated that using local context does help GraphST perform better than with
 the global context. We have added these results to Supplementary Figure S14A in the revised
 manuscript.

Moreover, GraphST is also different from DGI in their contrastive loss functions. DGI uses single
 contrastive loss, while GraphST employs symmetric contrastive loss by adding a contrastive corrupted
 loss term (formula (5)). Symmetric contrastive loss can help stabilize the model and learn a better
 representation. To demonstrate the advantage of symmetric contrastive loss over single contrastive
 loss, we conducted an ablation study to compare GraphST with a variant that does not include
 contrastive corrupted loss. We tested GraphST and the variant on the 12 DLPFC samples and
 evaluated their performance with the ARI metric. Figure R5 shows that GraphST achieved a median
 ARI score of 0.60, an improvement of 28% compared to the variant (median ARI score of 0.47). Thus,
 we conclude that symmetric contrastive loss does improve the model's performance. We have added
 the results in Supplementary Figure S14B of the revised manuscript.

**Comment 1.4.** What are the meaning and motivation of formula (5), line 734? It's important to show the
 performance difference with and without adding this term by experiment, because DGI only contains (4)
 instead of (5). Does this term actually improve the performance?

**Response 1.4:** Thank you very much for your question and suggestion. Formula (5) is a contrastive
 corrupted loss function that is symmetric to formula (4). The combination of loss functions (4) and (5)
 forms a symmetric contrastive loss that can make GraphST's model training more stable and robust,
 thus improving the spatial clustering performance.

As shown in our GraphST workflow (Figure R6), we use the original graph as input to create a corrupted
 graph by randomly shuffling features across spots while keeping the adjacency matrix of the graph
 unchanged. The original and corrupted graphs are thus structurally identical. At first, we followed DGI
 in constructing the contrastive learning with only a single contrastive loss (i.e., formula (4) in the revised
 manuscript). However, during model training, we found that the loss curve was unstable, as shown in
 Figure R7. Motivated by that and the fact that the original and corrupted graphs are structurally
 symmetric, we added a symmetric (corrupted) contrastive loss function to make the model training more
 stable and robust.

Based on your comments, we conducted an ablation study to validate the effectiveness of symmetric
 contrastive loss. We compared GraphST with a variant without contrastive corrupted loss on the 12
 DLPFC samples and used the ARI metric for evaluation. The results in Figure R5 show that GraphST
 without contrastive corrupted loss achieved a lower median ARI score (0.47) than the original GraphST
 (0.60), supporting the idea that symmetric contrastive loss helps our model achieve better performance.
 Furthermore, Figure R7 shows that the model training curve is stabilized with symmetric contrastive
 loss. These results have been included in Figure S14B in the Supplementary.

Figure R6. Workflow of GraphST for spatial clustering.

Figure R7. Training loss curves with and without contrastive corrupted loss on four DLPFC samples.

**Comment 1.5.** The reason of using reconstructed expression data to cluster instead of using the latent
 embedding need to be justified. Moreover, why choosing mclust over graph-based methods such as
 Leiden, Louvain? It's important to justify such choices in terms of data analysis and results.

**Response 1.5:** Thank you very much for your very constructive comments. In our framework, the
 reconstructed expression is more informative than the latent representation for two reasons. Firstly, as
 shown in Figure R6, our GraphST framework consists of a GCN (graph convolutional network)-based
 encoder and a GCN-based decoder. The encoder and decoder have symmetrical structures and equal
 numbers of GCN layers. In our model, the number of layers for encoder and decoder are set to 1. The
 basic principle of GCN is to update the node representation by iteratively aggregating information from
 the neighbours. Therefore, as the output of the decoder, the reconstructed expression contains more
 information about the local context than the latent representation, as the reconstructed expression
 aggregates feature information of two-hop neighbours while the latent representation only aggregates
 one-hop neighbours' feature information. Secondly, compared to the latent representation, the
 reconstructed expression captures more topological structure and semantic information. This is
 because the reconstructed expression is obtained through two GCN layers, meaning that the adjacency
 matrix is used twice.

In response to your comments, we compared the clustering performance of using the latent
 representation and the reconstructed expression on the 12 DLPFC samples. The results in Figure R8
 show that GraphST achieved much a higher median ARI score when using the reconstructed
 expression for clustering than the latent representation, suggesting that the former contains more useful
 information than the latter. These results have been included in Supplementary Figure S14D.

Figure R8. Comparison analysis between latent representation and reconstructed expression on
 DLPFC.

We chose mclust as the default clustering method because our assessment showed that mclust
 performs better than Leiden and Louvain in most cases. Figure R9 shows the clustering results on the
 12 DLPFC samples using Leiden, Louvain, and mclust. mclust consistently outperformed Leiden and
 Louvain on all 12 samples in terms of the ARI metric, with a much higher median ARI score (Figure
 R10). Visually, the clusters identified by mclust are more continuous. These results have been included
 in Supplementary Figure S15. Here, we would like to mention that several previously published spatial
 clustering methods such as BayesSpace (Zhao et al., 2021) and STAGATE (Dong and Zhang, 2022)
 also use mclust. Nevertheless, we have now added Leiden and Louvain to GraphST as alternative
 options.

**Reference**

Zhao et al. Spatial transcriptomics at subspot resolution with BayesSpace. Nature Biotechnology,
 39(11), 1375-1384.

Kangning Dong and Shihua Zhang. Deciphering spatial domains from spatially resolved transcriptomics
 with an adaptive graph attention auto-encoder. Nature Communications. 2022.

Figure R9. Comparison analysis between Leiden, Louvain, and mclust with the output of GraphST as
 input on DLPFC.

Figure R10. Boxplots of clustering accuracy of Leiden, Louvain, and mclust with the output of GraphST
 as input on DLPFC in terms of ARI.

**Comment 1.6.** *In line 684, do the authors augment data through creating corrupted graph by randomly*
 *adding or dropping edges? What is the effect of such procedure on the overall performance of the*
 *method.*

**Response 1.6:** We are sorry for the confusion. In our framework, instead of randomly adding or
 dropping edges, we create a corrupted graph by randomly shuffling gene expression vectors between
 spots while leaving the adjacency matrix of the graph unchanged. In contrastive learning, data
 augmentation aims to increase the diversity of training data and thus enhance the model's learning
 capability. Therefore, the data augmentation procedure plays an important role in contrastive learning.
 During data augmentation, the distribution of the augmented data should be distinguishable from the
 original data. Otherwise, it can easily cause the model to overfit. Therefore, we adopted random feature
 swapping to perturb the original data as much as possible, such that the model can learn more useful
 information from the spatial data.

**Comment 1.7.** *Regarding the data integration performance shown in Figure 4, why did the authors not*
 *compare many other methods designed for nonspatial scRNA-seq data, such as scVI (Lopez et al. 2018)*
 *and Harmony (Korsunsky et al. 2019), because those classical methods have been well demonstrated*
 *for good performance for single-cell data.*

**Response 1.7:** Thank you for your insightful comments. Following your comments, we added scVI and
 Harmony to our tests on the two mouse breast cancer datasets for vertical integration (Figure R11).
 Both scVI and Harmony were able to mix the two slices, but some batch differences were still visible
 post integration (Figure R11B). In comparison, GraphST evenly mixed the two slices, achieving better
 batch mixing than scVI and Harmony. We also quantitatively evaluated batch mixing with the iLISI metric
 where the higher iLISI score, the better the batch mixing. GraphST achieved a much higher iLISI score
 than Harmony and scVI (Figure R11C), confirming our visual observations. In the post-integration
 clustering, Harmony failed to align clusters across the two slices. scVI performed better than Harmony
 but some clusters were still not accurately mapped, such as clusters 1, 5, and 10. In contrast, GraphST's
 clusters highly overlapped between the two slices.

We also tested scVI and Harmony on one more mouse breast cancer sample (Figure R11D-F). Batch
 differences remained visible on the Harmony UMAP plot (Figure R11E). Comparatively, scVI removed
 the batch effects much better than Harmony. GraphST performed the best by evenly mixing the two
 slices. In terms of iLISI, Harmony significantly underperformed GraphST while scVI was comparable to
 GraphST (Figure R11F). Most of Harmony's clusters did not match across the two slices. While scVI
 generated clusters that were more consistent than Harmony, some clusters were fragmented,
 especially in section 2. GraphST again identified clusters that were spatially coherent and aligned well
 across the two slices. These results have been added to Figure 4 of the revised manuscript.

Figure R11. Vertical integration results of different methods on the two mouse breast cancer datasets.

**Comment 1.8.** It's important to show the STAGATE results that similar to Fig 4E to better demonstrate
the data integration performance.

**Response 1.8:** Thank you for your great suggestions. Following your suggestions, we tested the
performance of STAGATE on the two mouse brain samples. For fair comparison, we set the same
number of clusters for all methods, i.e., 26 clusters. As shown in Figure R12, both GraphST and
STAGATE produced continuous clusters that match the Allen brain reference well. Most importantly,
like GraphST, the STAGATE's clusters were aligned along the edges of the anterior and posterior
sections. However, some key brain regions were not represented in STAGATE's clusters. For example,
STAGATE failed to identify the dorsal (top) and ventral (bottom) horn of the hippocampus regions
highlighted with white boxes on the H&E images. In contrast, GraphST was able to reveal these regions.
Overall, compared with STAGATE, GraphST performed slightly better in the horizontal integration task.
We have added STAGATE's results to Figure 4 of the revised manuscript.

Figure R12. Horizontal integration results of different methods on two mouse brain samples.

**Minor points:**

**Comment 1.9.** To better support the manual annotation result in Fig 6A, the spatial expression
 distribution of several marker genes for each panel in Fig 6A need to be added.

**Response 1.9:** The breast cancer tissue was ER positive, PR negative, Her2 positive, and diagnosed
 with ductal carcinoma in situ, lobular carcinoma in situ, and invasive carcinoma. Our pathologist
 collaborator produced the manual annotation based on the H&E image. Morphologically, it is easier
 to discern the IDC, DCIS, healthy, and tumour edge regions. As tumours usually harbour high cellular
 heterogeneity, it is challenging to find known gene expression markers that can distinguish IDC from
 DCIS. Here, we selected several reported marker genes of breast cancer from literature and plotted
 their spatial expression to support our manual annotation (Figure R13). We have added the marker
 genes to Supplementary Figure S13.

**Reference**

The Cancer Genome Atlas Network. Comprehensive molecular portraits of human breast tumours,
 Nature 2012.

Figure R13. Spatial expression distribution of reported breast cancer markers

**Comment 1.10.** *In line 836, the author mentioned the first loss term indicates contrastive loss, why is*
*there only one instead of two terms? What is the meaning of the first term?*

**Response 1.10:** Thank you for your comments. Formula (10) is the overall objective function of
GraphST' third module for cell type deconvolution of ST data. In our framework, we use two different
contrastive learning methods for the first and third modules, respectively. Motivated by Zhang et al.
(2022), we use an augmentation-free contrastive learning method for the third module. Therefore, there
is only one term for contrastive loss. The first term (i.e., contrastive loss) in formula (10) is an InfoNCE
objective function that aims to maximize the similarities of positive pairs and minimize those of negative
pairs. We have added more details in the revised manuscript to describe the contrastive learning
method of the third module.

**Reference**

Zhang et al. Dual Temperature Helps Contrastive Learning Without Many Negative Samples:
Towards Understanding and Simplifying MoCo. CVPR2022.

**Comment 1.11.** *In line 712, which norm is used? L1 or L2 or others?*

**Response 1.11:** Thank you for your comments. The norm term in formula (3) is L2-norm. We have
revised formula (3) in the revised manuscript.

**Comment 1.12.** *In Fig 3C, the titles of panels Mesenchyme and Dermomyotome seem misplaced.*

**Response 1.12:** Thank you very much for pointing out this. We have changed the titles of these two
panels. Please refer to Figure 3C in the revised manuscript.

**Comment 1.13.** *All color bars need to be explained for their meanings.*

**Response 1.13:** Thank you for your comments. We have added legends to all colour bars in the revised
manuscript.

**Comment 1.14.** *Many typos and grammar errors in the manuscript, e.g., in line 28, "has" should be*
*"have"; in line 59, "K-means" should be "k-means"*

**Response 1.14:** Thank you very much for your careful reading. We have carefully gone through our
manuscript and corrected the typographical and grammatical errors.

**Comment 1.15.** *Lines 124-126, "Using self-supervised contrastive learning improves performance in*

*learning relevant latent features and has the additional benefit of removing batch effects”. This sentence*
*occurs without any supporting evidence. It needs to be fixed.*

**Response 1.15:** Thank you very much for pointing out this. We have revised this sentence as follows.

*“Using self-supervised contrastive learning improves performance in learning relevant latent features.”*

To demonstrate the contribution of self-supervised contrastive learning, we conducted an ablation study
by comparing GraphST with a variant of GraphST without contrastive loss on the DLPFC dataset.
Without contrastive loss, the performance of GraphST is significantly reduced (Figure R14), indicating
that contrastive loss contributes to the performance improvement of our GraphST model. These results
have been added to Supplementary Figure S14C.

Figure R14. Comparison between GraphST and its variant GraphST without contrastive loss on the
DLPFC dataset.

**Comment 1.16.** *In lines 158-162, the authors introduced “self-reconstruction loss” and “contrastive loss”*
*and their effects. It’s important to show what the two losses are in the context of biology.*

**Response 1.16:** Thank you very much for highlighting this point. In our GraphST framework, we take
the gene expression matrix and spatial graph as inputs. The gene expression matrix contains the
feature information of spots while the spatial graph stores the spatial adjacency of spots. Our model is
a GNN-based model that aims to integrate the gene expression of spots with their corresponding spatial
information for spatial clustering. The key feature information is in the gene expression matrix which
should be retained. Therefore, we design the self-reconstruction loss to enforce the preservation of the
original gene expression information in the reconstructed expression.

The contrastive loss design is based on the assumption that a spot in the spatial data usually has a cell
type label similar to its local context, e.g., one-hop or two-hop neighbours. As discussed above in the
contrastive learning part, we define positive/negative pairs by pairing spot embedding h_i/h'_i from the
original/corrupted graph with its local summary vector s_{local} . The local summary vector s_{local} represents
the local context of a spot and is obtained by a sigmoid of the mean of all its neighbours’ embeddings.
The main goal of contrastive learning is to make spot embedding h_i close to its local context s_{local}
from the original graph. Therefore, trained with contrastive loss, spatially adjacent spots will have similar
embeddings while non-adjacent spots will have dissimilar embeddings.

Based on your comment, we have added more biological contexts when describing self-reconstruction
loss and contrastive loss in the revised manuscript.

**Comment 1.17.** *It’s unclear how the neighbor graph is constructed. In the caption of Fig. 1 (lines 1034-*
*1035), the authors wrote “...neighbor graph constructed using spot coordinates (x,y) of that fall within a*
*distance threshold”. However, in the method section in lines 665-675, the authors wrote “Finally, we*
*select the top k-nearest spots as its neighbors”. It’s unclear whether the authors used a distance*
*threshold or a threshold for k.*

**Response 1.17:** We are sorry for the confusion. In our framework, we use a threshold for k when
constructing the neighbourhood graph. We have carefully gone through the manuscript and ensured
consistency in the revised manuscript.

**Comment 1.18.** *Regarding the method (lines 655-659), the descriptions seem to be for the spatial*
*transcriptomics data. However, this is not clear from the description, as two kinds of datasets (spatial*
*transcriptomics data and scRNA-seq data) are mentioned in this paper.*

**Response 1.18:** Thank you for your comments. GraphST was developed for three analysis tasks,
spatial clustering, multiple ST data integration, and cell type deconvolution of spatial data. Cell type
deconvolution is achieved by projecting single-cell RNA-seq data onto the spatial data. Therefore, in
addition to spatial data, single cell RNA-seq data is also used in the third module of our framework.

**Comment 1.19.** *In lines 684-688, "...while keeping the original graph structure unchanged": was the*
*corrupted neighbor graph G' the same as the original G ?*

**Response 1.19:** Thank you for your comments. Yes, the adjacency matrix of the corrupted
neighbourhood graph G' is the same as the original G . When creating the corrupted neighbourhood
graph G' , we only randomly shuffle feature vectors between spots while keeping the graph's topological
structure unchanged. For example, the entire feature vector of node A is assigned to node B and vice
versa.

**Comment 1.20.** *In lines 708-709, " W_d and b_d represent the trainable weight matrix and bias vector,*
*respectively, which are shared by all nodes in the graph". Please justify why W_d and b_d need to be*
*shared by all nodes in the graph. Besides, is this the same case for W_e and b_e ?*

**Response 1.20:** Thank you very much for your insightful comments. In our model, without loss of
generality, the trainable weight matrices W_e, W_d and bias vectors b_e, b_d are shared by all nodes in the
graph. In the GNN model, the dimensions of the weight matrices and bias vectors are usually very large
depending on the number of nodes of the input graph. For example, in the datasets used in our
manuscript, the smallest (i.e., DLPFC slice #151676) has 3460 spots and the largest (i.e., Mouse
embryo E14.5) has 92,928 bins. If we use a different weight matrix and bias vector for each node, it will
be very challenging to train the model. Sharing the weight and bias significantly reduce the number of
weight and bias terms used, making it easier to train the model. It also helps reduce the running time.

**Reference**

Korsunsky, Ilya, Nghia Millard, Jean Fan, Kamil Slowikowski, Fan Zhang, Kevin Wei, Yuriy Baglaenko,
Michael Brenner, Po-Ru Loh, and Soumya Raychaudhuri. 2019. "Fast, Sensitive and Accurate
Integration of Single-Cell Data with Harmony." *Nature Methods* 16 (12): 1289–96.
Lopez, Romain, Jeffrey Regier, Michael B. Cole, Michael I. Jordan, and Nir Yosef. 2018. "Deep
Generative Modeling for Single-Cell Transcriptomics." *Nature Methods* 15 (12): 1053–58.
Veličković, Petar, William Fedus, William L. Hamilton, Pietro Liò, Yoshua Bengio, and R. Devon Hjelm.
2018. "Deep Graph Infomax." *ArXiv [Stat.ML]*. arXiv. <http://arxiv.org/abs/1809.10341>.

**Reviewer #2** (Remarks to the Author):

*In the manuscript "GraphST: A novel graph" by Long and coworkers the authors develop a new*
*method for better describing spatial transcriptomics data and being able to integrate multiple studies.*

*The method is based on graph neural networks and contrastive learning, which makes it possible to*
*combine scRNA-seq of better resolution and spatial transcriptomics. The method is logically sound and*
*makes a lot of sense. Moreover, authors show that it empirically identifies more relevant clusters and*

*allows data integration for higher power. Although, I am not an expert in spatial transcriptomics these*
*problems seem of great importance and authors spend good effort to show that it works a planned.*

*Having said that my expertise is in neural networks and translational bioinformatics I believe that the*
*paper would be a good contribution to the spatial transcriptomics field. From my side, I have no concerns*
*of the paper and like to see it published.*

**Response:** We thank the reviewer for the above positive comments. We welcome you to provide
additional valuable comments.

Reviewers' Comments:

Reviewer #1:

Remarks to the Author:

The revision has fully addressed my comments and concerns. Well done.